# Non-diffusive slow heat dissipation induces high local temperature in living cells

Masaharu Takarada [1], Ryo Shirakashi [2], Masahiro Takinoue [3,4,5], Motohiko Ishida[1], Masamune Morita[3,6], Hiroyuki Noji [7], Kazuhito V. Tabata [4,7,8], Takashi Funatsu [1,9] & Kohki Okabe [1,4] ✉

Recently, intracellular thermometry has revealed temperature variations within cells. Although the biological significance of intracellular temperature change is recognized, the physical principles of intracellular temperature change remain a mystery. Here, we investigate intracellular heat transfer through intracellular temperature mapping using a fluorescent polymeric thermometer and high-speed fluorescence lifetime imaging microscopy. Through infrared laser irradiation-assisted heating, we track changes in temperature distribution to examine the mechanism of intracellular heat dissipation in comparison with heat conduction. Continuous heating induces the significantly slower relaxation of the average temperature of single cells compared with that of liposomes containing homogeneous aqueous solutions of comparable size; to the scale of seconds. We additionally elucidate that these phenomena are impacted by intracellular structures and molecules. Finally, we discover that this slow intracellular temperature relaxation originates from non-diffusive heat dissipation distinct from the conventional heat conduction model. Our results provide insights into the mechanisms of temperature variation in cells that are unresolved based on our current understanding, establishing a framework for understanding intracellular thermodynamics under non-equilibrium conditions.

Temperature has a great influence on biological activity at various hierarchical levels. It is well established that physiological functions include adapting to and utilizing changes in ambient temperature. The study of the control of biological functions through heating has a long history, under topics such as hyperthermia therapy in medicine and the heat shock response in biology. In recent years, intracellular temperature has attracted increasing attention. Fluorescent molecular thermometers (fluorescent polymeric thermometers (FPT)[1,2], fluorescent proteins[3–5], fluorescent small molecules[6], and

nanoparticles, such as upconversion nanoparticles[7,8] and fluorescent nanodiamonds[9–11]) and nonoptical methods (thermocouples[12,13] and electrochemical methods[14]) have revealed temporal and spatial variations of temperature within cells. These intracellular temperature changes are characterized by spontaneous heat generation within cells and are associated with cellular functions[15–18].

Very recently, it has been reported that heat endogenously generated within cells in response to physiological stimuli drives cellular functions, such as neuronal differentiation[9], brain edema[19],

[1]Graduate School of Pharmaceutical Sciences, The University of Tokyo, Bunkyo-ku, Japan. [2]Institute of Industrial Science, The University of Tokyo, Meguro-ku, Japan. [3]Laboratory for Chemistry and Life Science, Institute of Science Tokyo, Yokohama, Japan. [4]JST, PRESTO, Kawaguchi, Japan. [5]Research Center for Autonomous Systems Materialogy (ASMat), Institute of Science Tokyo, Yokohama, Japan. [6]Molecular Biosystems Research Institute, National Institute of Advanced Industrial Science and Technology (AIST), Tsukuba, Japan. [7]Graduate School of Engineering, The University of Tokyo, Bunkyo-ku, Japan. [8]Research Institute of Planetary Health, The University of Tokyo, Minato-ku, Japan. [9]Graduate School of Integrated Sciences for Life, Hiroshima University, Higashi-Hiroshima, Japan. ✉e-mail: okabe@mol.f.u-tokyo.ac.jp

and heat shock response[20]. In these phenomena, the driving force of the functions is not a change in external environmental temperature, but a temperature variation of about 1–2 °C induced by spontaneous intracellular heat. This emerging phenomenon produced by intracellular heat propagation (termed thermal signaling) is considered a potential universal intracellular signaling mechanism as all molecular dynamics and chemical reactions are governed by heat. Thus, intrinsic temperature changes in cells are a promising unexplored property in biology and medicine that drives cellular activity and determines the cellular state[15–18].

Although the biological significance of intracellular temperature variations is being revealed, the mechanism by which heat generated within the cell produces this temperature change of approximately 1–2 °C remains unknown. It has been pointed out that, according to the heat conduction equation, temperature changes due to spontaneous heat generation within cells can only reach approximately $10^{-5}$ °C (the $10^5$ gap issue)[21]. While the values calculated from the heat conduction equation and experimental observations show a significant discrepancy, the observed result of intracellular temperature change by about 1 °C due to spontaneous heat generation has been verified through robust reproducibility, across different methodological principles[15]. Via attempts to explain the physical mechanism behind this discrepancy, it has been proposed that heat generated within intracellular regions where heat conduction is restricted could lead to elevated temperatures[5,22,23]. Recently, various studies have estimated cellular thermal conductivity (0.1–0.6 W m$^{-1}$ K$^{-1}$)[23–26] and thermal diffusivity (0.27–1.34 × 10$^{-7}$ m$^2$ s$^{-1}$)[5,27] to be between one-sixth and one-fold of those in water. It should be noted that the actual thermal conductivity and diffusivity reported in these studies showed significant variation. Even assuming conditions in which heat conduction is severely restricted (thermal conductivity: 0.1 W m$^{-1}$ K$^{-1}$, thermal diffusivity: 0.27 × 10$^{-7}$ m$^2$ s$^{-1}$), the discrepancy between the experimental results and the calculated values remains substantial (over $10^4$-fold). This gap cannot be fully explained by considering additional parameters, such as the intensity of and the distance from the heat source. Thus, this issue concerning intracellular temperature variation is not a matter of the cellular thermal properties, but rather a fundamental mystery of the heat dissipation process itself[18]. This unresolved paradox of intracellular thermodynamics hinders the physical and biochemical understanding of the mechanisms of all chemical reactions and thermal signaling phenomena in cells. Despite the potential benefits of addressing this challenge, which could open new fields in cell biology and biophysics, no experimental investigation of the fate of intracellular heat has been performed to date.

In this study, in order to perform a detailed analysis of the dissipation process of heat generated in cells, we observed the heat transfer using a controllable heat source and a method for tracking the changes in the temperature distribution within the cells. First, we developed a real-time temperature mapping method using FPT by utilizing a time-correlated single-photon counting (TCSPC)-based FLIM instrument, which has a very high photon acquisition capacity, in addition to employing a method for highly efficient fluorescence lifetime determination. Next, we used the developed real-time temperature mapping method to track the transient temperature change caused by an artificial heat source in living cells and liposomes, revealing that the temperature relaxation in cells is slow and, moreover, that it is location- and molecule-dependent inside of the cells. The detailed analysis of the intracellular temperature relaxation performed in comparison with the heat conduction equation revealed that the intracellular temperature changes during continuous heating, similar to physiological thermogenesis, include temperature changes that do not depend on heat conduction. These results suggested that

thermal energy remains localized without immediate dissipation, and the resultant local heat controls thermal signaling in cellular functions.

## Results

### High-speed temperature mapping in living cells

In this study, to explore the process of the change in temperature distribution within cells, we adopted FPT[2] (Fig. 1a), which responds sensitively to temperature changes above the lower critical solution temperature. Intracellularly introduced FPT exists in a dispersed state within cells and can capture the temperature distribution[2]. This ability is advantageous over other intracellular temperature measurement methods, such as fixed-point temperature measurement using fluorescent nanoparticles or temperature tracking within specific organelles using fluorescent small molecules and proteins. We previously demonstrated that FPT selectively detects intracellular temperature changes due to its functional independence; FPT did not respond to physicochemical factors other than temperature, such as pH, ionic strength, biomolecular concentration, and viscosity, when they were quantitatively altered in solution[2]. In this study, we used a control copolymer (CP)[2] with a chemical composition almost identical to that of FPT, but lacking temperature sensitivity, to confirm the temperature selectivity of FPT. In complex environments like the intracellular space, various physicochemical changes can occur due to heating. Therefore, rigorous verification using this CP is essential to achieving our objectives. The inclusion of a control probe that enables the investigation of temperature selectivity is a unique feature of FPT that is not found in other nanothermometers.

To achieve the spatio-temporal resolution of temperature measurements that allows tracking of intracellular heat, we changed the method for calculating fluorescence lifetime in TCSPC. Conventional fluorescence lifetime determination via exponential approximation requires a large number of photons, such that a 60-s signal integration barely provides a snapshot of the temperature distribution with organelle-sized spatial resolution[2]. Because we expected linear approximation to reduce variability in fluorescence lifetime determination, we adopted the fast fluorescence lifetime[28] defined by the average photon arrival time ("fast lifetime"), which requires fewer photons than the conventional method (Fig. 1b and Supplementary Note 1).

First, we compared the fast lifetime and conventional fluorescence lifetime of FPT (Fig. 1a, c, and Supplementary Fig. 1). Figure 1c shows the relationship between the temperature and the fast lifetime of FPT in the solution of COS7 cell extracts. The temperature response curve of FPT by fast lifetime was almost the same as that of the conventional method[2]. This difference in the fluorescence lifetime determination method also did not affect the temperature response curve using living cells (Supplementary Fig. 1). We confirmed that CP does not exhibit temperature dependence even in fast lifetime measurements (Fig. 1d).

We investigated whether reliable fast lifetime measurements could be achieved using fewer photons than with conventional methods. We experimentally evaluated the impact of different fluorescence lifetime estimation algorithms on the trueness and precision of the resulting values. (Fig. 1e, f, and Supplementary Fig. 2). The results showed that fast lifetime-based fluorescence lifetime determination outperformed conventional methods in terms of trueness (Fig. 1e) and precision (Fig. 1f and Supplementary Fig. 2). Furthermore, it was confirmed that the errors and variability associated with FPT fast lifetime determination using a limited number of photons did not significantly affect the measurement values in this study (Supplementary Note 1).

Here, we performed intracellular temperature mapping with fast lifetime of FPT in living COS7 cells using the state-of-the-art FLIM having high photon acquisition capability (photon detection efficiency

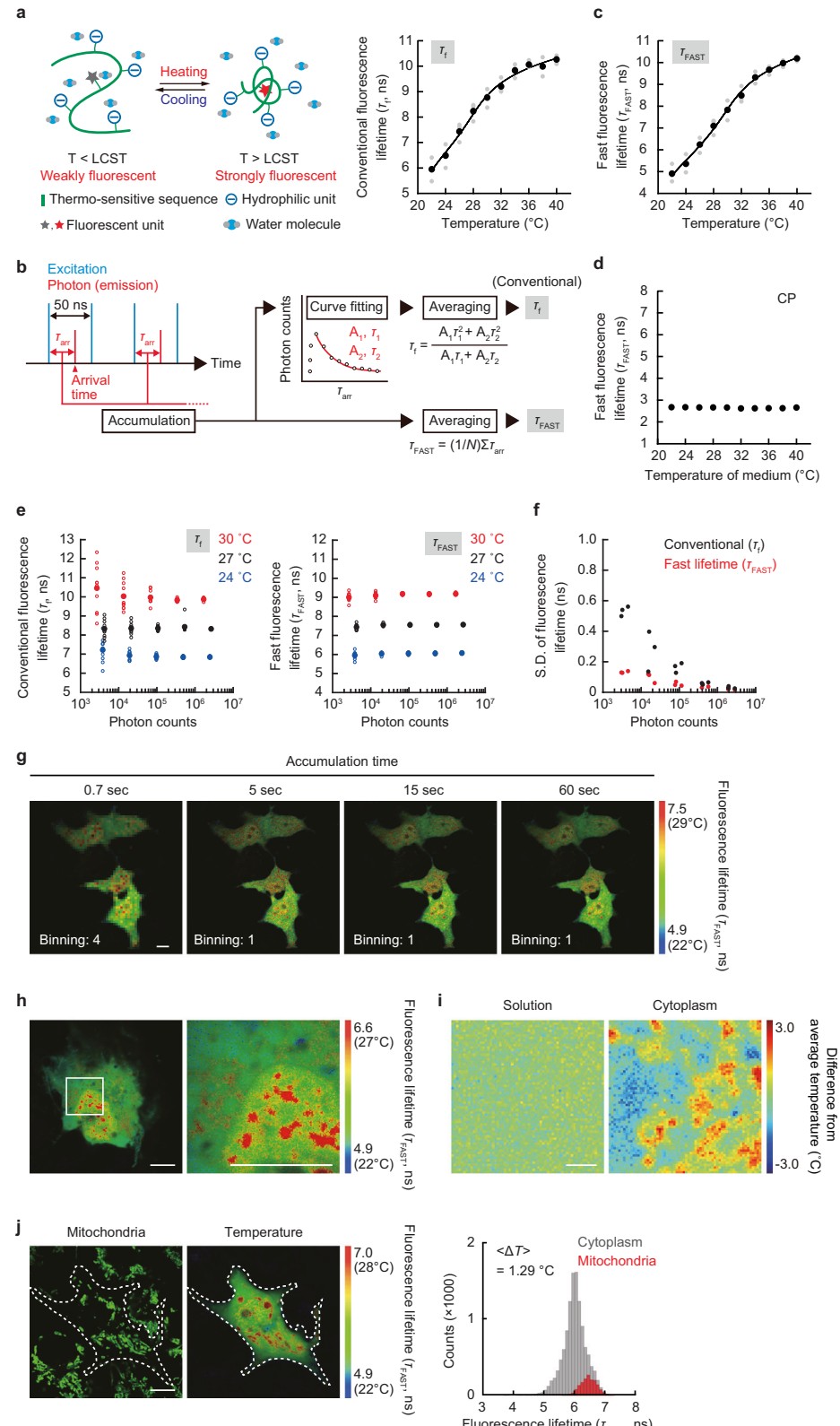

per unit time is 25 times higher than that of our previous system[2]). Comparing the fluorescence lifetime images at different measurement times, we found that the intracellular temperature mapping became more detailed in a time-dependent manner; the spatial heterogeneity of the intracellular temperature could be observed even at a measurement time of only 0.66 s, while a previous study using FPT required a measurement time of 60 s (Fig. 1g)[2]. The highly efficient

fluorescence lifetime determination of this method enabled fluorescence lifetime imaging at high compartmentalization (e.g., 512 × 512 pixels). The results of high-resolution temperature mapping in a single COS7 cell (Fig. 1h) clearly revealed a heterogeneous temperature distribution not only at the organelle level, such as a high temperature in the nucleus, but also in micro-spaces in the cytoplasm and nucleoplasm. When the heterogeneous temperature distribution visualized in

**Fig. 1 | High-resolution mapping of temperature in living COS7 cell with $\tau_{FAST}$-based FLIM. a** Functional diagram of FPT responding above the lower critical solution temperature (LCST) and its temperature response curve using conventional fluorescence lifetime ($\tau_f$). **b** Schematic diagram of fluorescence lifetime determination in FLIM. In the conventional analysis, the fluorescence lifetime ($\tau_f$) is calculated from the parameters obtained by approximating the fluorescence decay curve with a two-component exponential function, while the fast fluorescence lifetime ($\tau_{FAST}$) is obtained by just averaging the arrival time ($\tau_{arr}$) of photons in TCSPC. A and N represent amplitude and number of photons, respectively. **c** The temperature response curve of FPT using fast fluorescence lifetime ($\tau_{FAST}$). **d** Relationship between temperature and the fast fluorescence lifetime ($\tau_{FAST}$) of the control copolymer (CP). **e** Relationship between the accumulated photon count and fluorescence lifetime at 24, 27, and 30 °C ($\tau_f$: left and $\tau_{FAST}$: right). **f** The relationship between photon counts in TCSPC-FLIM and the standard deviation of fluorescence lifetime ($\tau_f$: black and $\tau_{FAST}$: red). **g** Comparison of fluorescence lifetime images of FPT using $\tau_{FAST}$ with various accumulation times. **h** Fluorescence lifetime ($\tau_{FAST}$) images of FPT. The region of interest, as shown in the square in the left panel, is enlarged in the right panel. **i** Comparison of temperature distribution in aqueous solution and in the cytoplasm of a cell. Scale bar represents 2 μm. **j** The colocalization of the intracellular area with a higher temperature than the surroundings (middle) with mitochondria visualized using MitoTracker Deep Red FM (left). The histogram of fluorescence lifetime ($\tau_{FAST}$) is shown on the right. Dotted line indicates the outline of the cell. $\Delta T$ was calculated by subtracting the average temperature of the cytoplasm from that around mitochondria. <$\Delta T$> represents an average of the histogram. The experiments (**g**–**j**) were performed three times with similar results. Scale bars in **g**, **h**, **j** represent 10 μm. Source data are provided as a Source data file.

an intracellular micro-space was compared with that in a homogeneous aqueous solution of the same size, the spatial fluctuation of the intracellular temperature was notably large (Fig. 1i). Next, mapping of the temperature distribution was performed in cells in which mitochondria, the main heat-generating organelles, were visualized (Fig. 1j). As in our previous study[2], the mitochondria were clearly shown to have a higher temperature ($\Delta T = 1.29$ °C) than the surrounding region in steady-state cells.

## Local heating using IR laser in living cells

Next, to observe the change in intracellular temperature with this mapping method, we heated an intracellular local area via infrared (IR) laser (1480 nm) irradiation as an artificial heat source. By directly stimulating the O−H vibration of water molecules, an irradiating IR laser focused on a glass surface (Fig. 2a) can heat up an aqueous solution in a local area (about 1 μm in diameter) in a transient, and power-dependent manner[29,30]. The near-infrared spectrum of water showed absorption of light at the wavelength of the IR laser used in this study (1480 nm), while no absorption of this light by FPT itself was observed (Supplementary Fig. 3). Considering that the fluorescence lifetime values and the components of the fluorophore (7-[N,N-dimethylaminosulfonyl]-2,1,3-benzoxadiazol-4-yl, DBD) in FPT are sensitive to the chemical environment (e.g., hydrophobicity)[31], we analyzed the details of the fluorescence decay curve of FPT at the heating center in the nucleus and the cytoplasm. As shown in Supplementary Fig. 4, the concordance of the fluorescence decay curves (i.e., the fluorescence lifetimes and their compositions of the two components) during IR laser heating and medium heating was confirmed. Furthermore, the analysis of the fluorescence decay curve of CP confirmed that factors other than temperature (e.g., viscosity) induced by IR laser irradiation do not directly affect the FPT response (Supplementary Fig. 5). Figure 2b shows the results of fluorescence lifetime imaging in the cells under IR laser irradiation, demonstrating the concentric distribution of temperature around the point of IR laser irradiation. On the other hand, IR laser irradiation did not change the fluorescence lifetime distribution of CP (Fig. 2c). These results clearly demonstrate that the spatial distribution of the fluorescence lifetime of FPT created by IR laser irradiation is solely due to heat, independent of the chemical environment. When the nucleus was irradiated with the IR laser, the temperature gradient formed in the nucleus was steeper than that in the cytoplasm, even though the amount of IR laser power applied to the nucleus and the cytoplasm was the same (Supplementary Fig. 6). Next, the increase in intracellular averaged temperature upon IR laser irradiation was dependent on the laser intensity (Fig. 2d). To investigate the reliability of intracellular temperature change recorded during this artificial heating, we examined the results using another fluorescent thermometer with a completely different temperature response principle. Rhodamine B has long been known for the high temperature dependence of its fluorescence quantum yield[32] and has been used to measure solution temperature in nanoscale space[33]. The

fluorescence intensity-based measurement of temperature using fluorescent small molecules is not suitable for intracellular temperature mapping, observation of local temperature where local fluorophore concentration changes, or temperature measurement in systems where environmental factors may change. However, it can be used to track the average temperature of the entire cell because this study confirmed that the effects of IR laser irradiation on factors other than heat are negligible. Here, we quantitatively investigated the intracellular temperature change during IR laser irradiation using the temperature-dependent change in fluorescence intensity of intracellular Rhodamine B-labeled dextran (molecular weight 10,000) (Supplementary Fig. 7). The results showed that the change in averaged temperature during IR laser heating was the same between the two methods, which confirmed the validity of the temperature change measurement. In addition, by repeating the temperature mapping upon transient intracellular heating with an IR laser, we examined the ability to track the intracellular temperature distribution using this method. As shown in Fig. 2e, the increase in fluorescence lifetime upon IR laser irradiation returned to the pre-irradiation value when the irradiation was stopped, and the temperature gradient could be repeatedly formed. Lastly, we quantitatively verified the temporal and spatial resolution of this temperature measurement using an artificially temperature-manipulatable IR laser, and determined that they are 9 ms and at the diffraction limited level, respectively (Supplementary Figs. 8, 9, and Supplementary Note 2). These results clearly demonstrate that the spatial distribution of the fluorescence lifetime of FPT created by IR laser irradiation is solely due to heat, and that the resulting changes in intracellular temperature distribution can be tracked.

## Characteristics of intracellular temperature relaxation

By taking advantage of the high temporal resolution of this high-speed temperature mapping method (9 ms), we observed the time-course of temperature changes in cells during IR laser irradiation (Fig. 3a). Using CP, we confirmed that FPT responds only to temperature changes induced by IR laser irradiation (Fig. 3b). Here, the average temperature of a finite region containing the heat source was analyzed. This average temperature change represents the change in the thermal energy of this region, divided by the heat capacity. First, the changes in the averaged temperature of whole cells during pulsed and continuous heating are shown in Fig. 3c. The results of fast rise and fast relaxation of temperature change during pulsed heating confirm that our high-speed temperature mapping method is useful for tracking fast intracellular temperature change of several tens of milliseconds. The temperature relaxation during pulsed heating was consistent with the time resolution limit of the present temperature tracking method (Supplementary Fig. 8 and Supplementary Note 2). In the case of continuous heating, the temperature rapidly increased followed by a gradual increase upon IR laser irradiation, and then gradually decreased once irradiation was terminated (Fig. 3c). Because the

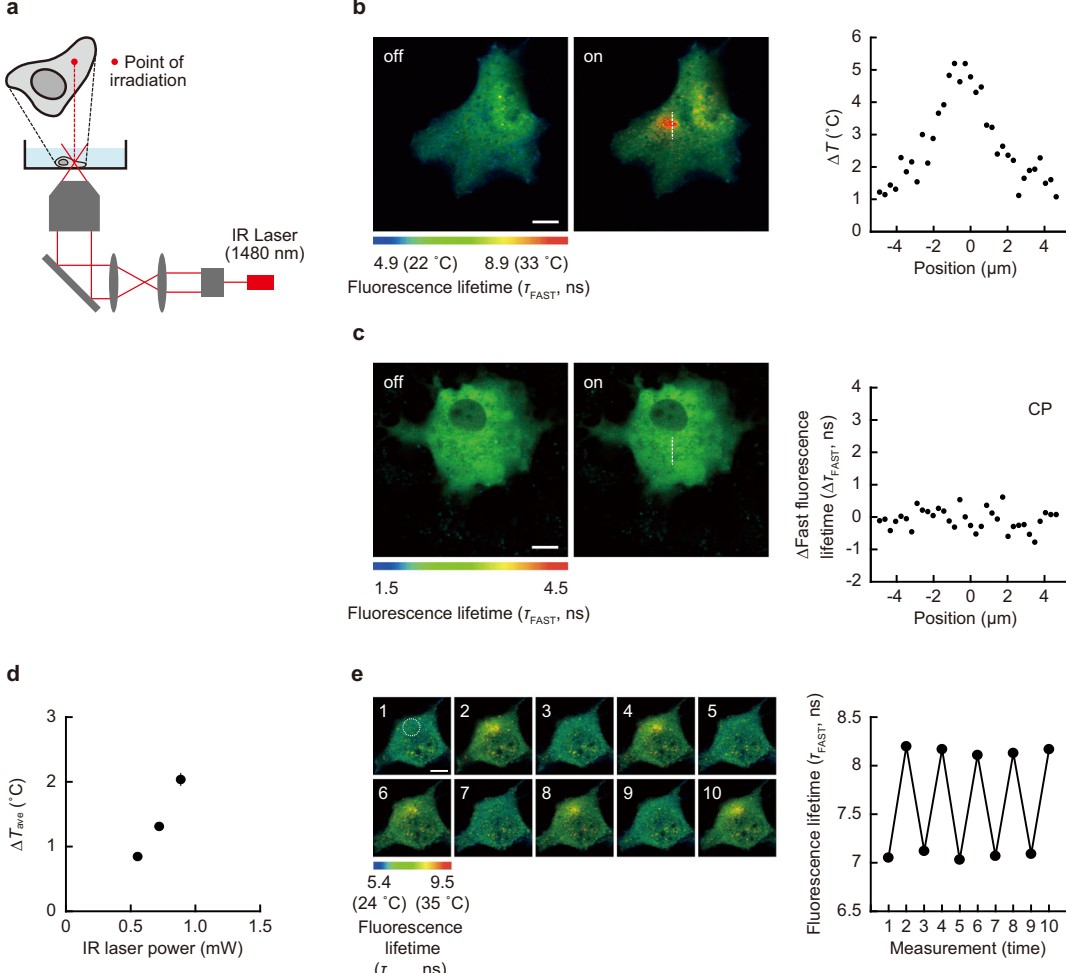

**Fig. 2 | Local heating using IR laser in living COS7 cell. a** Optical set-up. **b**, **c** Fluorescence lifetime images of FPT (**b**) or control copolymer (CP, **c**) without (off) and with (on) IR laser irradiation (0.55 mW). The right graph indicates the profile of $\Delta T$ (**b**) or $\Delta\tau_{FAST}$ (**c**) of the cell. **d** The relationship between IR laser power and the increase in averaged temperature of whole cells ($\Delta T_{ave}$). Data are presented as mean ± s.e.m. ($n = 30$ cells). **e** Reversibility of the fluorescence lifetime response of FPT to local heating (IR laser: 0.67 mW). The right graph shows the fluorescence lifetime of a 10 μm diameter area indicated by a dotted circle in the left image. The experiments (**b**, **c**, **e**) were performed three times with similar results. Source data are provided as a Source data file.

relaxation rate did not depend on the magnitude of the temperature rise (Supplementary Fig. 10), we normalized the temperature change and compared the curves of relaxation, finding that the relaxation after stopping continuous heating was slower than that in the case of pulse heating. As shown in Supplementary Note 3, the relaxation of the average whole-cell temperature after this continuous heating exhibited two-component time constants on the order of seconds. Further investigation of the heating duration dependence of the intracellular relaxation rate revealed that it slows down in a heating duration-dependent manner and gradually saturates after heating for more than 5–10 s (Supplementary Fig. 11). We also tracked the average temperature of whole cells using Rhodamine B. As shown in Supplementary Fig. 12, the slow intracellular temperature relaxation after continuous heating shown by Rhodamine B was in excellent agreement with the results obtained by FPT. In addition, we confirmed the slow intracellular temperature relaxation using Cy3 tagged by streptavidin (Supplementary Fig. 13). These results show that this slow temperature relaxation upon continuous heating was an intrinsic cellular property.

Tracking the temperature of local areas in cells can help us understand the principles that determine the organelle-specific temperature variation within them. Thus, we conducted temperature tracking of micro-spaces of cells. By following the changes in fluorescence lifetime in local regions of cells, we were able to track the

changes in average temperature of a finite region (a square area of 10 μm per side with vertical thickness of z-resolution in this imaging method) with a time resolution of several tens of milliseconds, and we observed the continuous heating-dependent slow temperature relaxation even in the local cytoplasm (Fig. 3d). Therefore, we compared the temperature relaxation in different locations in cells. We monitored the temperature in the nucleus when heated; as shown in Fig. 3e, the relaxation of the averaged temperature in the nucleus was slower than that in the cytoplasm. This slow nuclear temperature relaxation may be partly due to nuclear-specific transcription and RNA content, because transcription inhibition accelerated nuclear temperature relaxation (Supplementary Fig. 14). Furthermore, the rate of temperature relaxation in small spaces can be measured in giant plasma membrane vesicles (GPMVs) of 10 μm diameter directly isolated from living cells, which was significantly faster than a region of similar size in the cytoplasm (Supplementary Fig. 15).

## Cell-intrinsic temperature relaxation is significantly slower than heat conduction

To compare the results of slow temperature relaxation obtained in cells with a volume of water of the same size, we next attempted to track the temperature variations of liposome, a micrometer-sized artificial compartment with a lipid bilayer (Supplementary Fig. 16

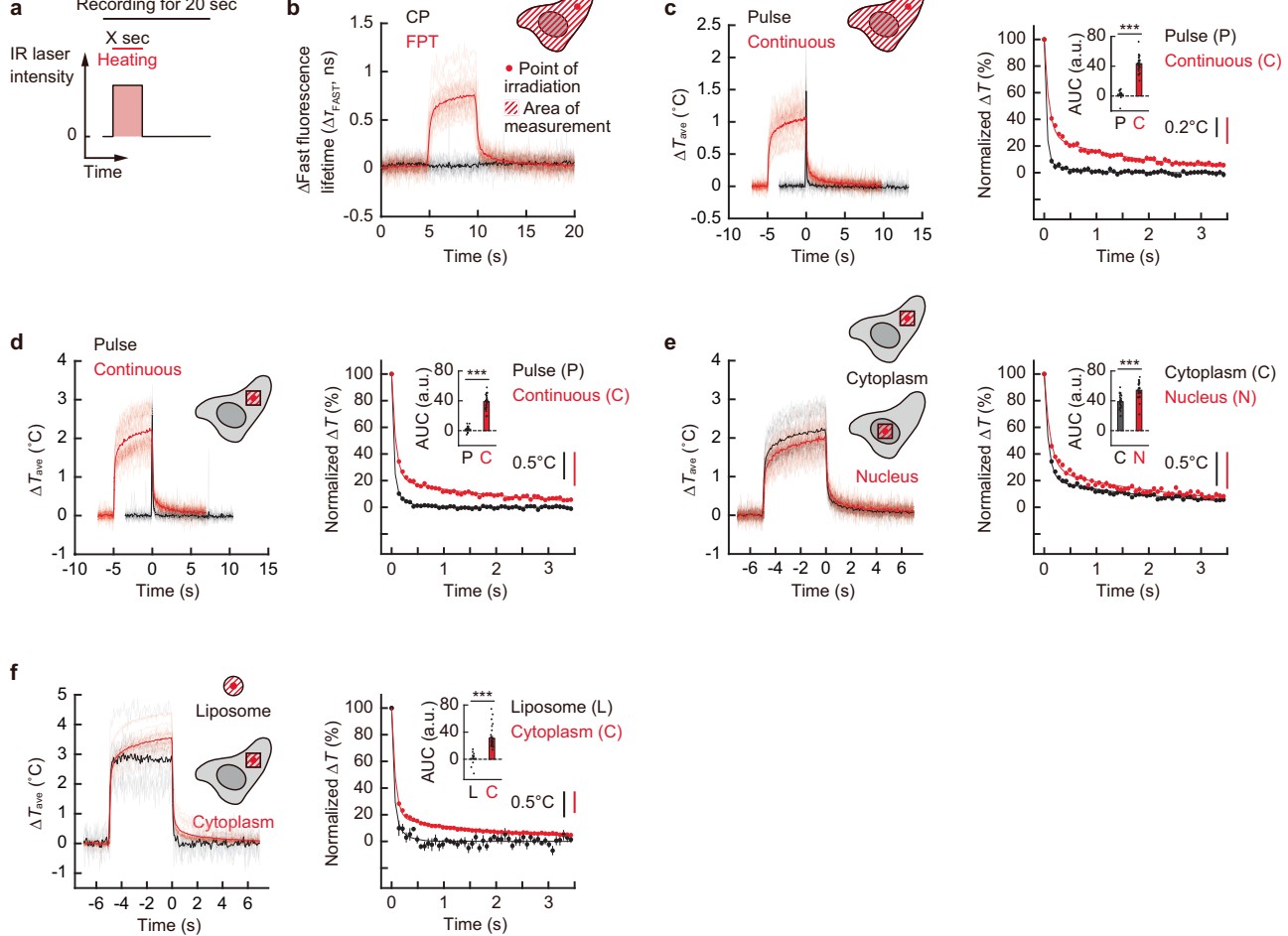

**Fig. 3 | Monitoring temperature variation upon heating with IR laser in living COS7 cell and liposome. a** Schematic diagram of transient heating. Heating was applied by an IR laser for X s during 20 s of intracellular temperature tracking. **b** Tracking of fast fluorescence lifetime changes of FPT (red) and control copolymer (CP, black) in cells when heated by continuous (0.55 mW for 5 s) irradiation with an IR laser (*n* = 29 [FPT] and 19 [CP] cells). **c–f** Tracking of the average temperature changes under IR laser heating and normalized temperature relaxation after heating stops. The averaged temperature of whole cells (**c**) and square area with a 10 μm side centered on the heating point (**d–f**) were analyzed. The results (**c, d**) of continuous heating (red, 0.55 mW, 5 s) are compared with those of pulsed heating (black, 1.1 mW, 0.1 s). The results (**e**) of continuous heating (0.55 mW, 5 s) of the nucleus (red) are compared with those of the cytoplasm (black). The results (**f**) of continuous heating (0.83 mW for 5 s) of the cytoplasm (red) are compared with those of water within liposomes. Data, including the inset in the right graph, are mean ± s.e.m. (Welch's two-sided t-test: **c**, *p* = 1.13 × 10⁻¹⁹; *n* = 29 [continuous] and 15 [pulsed] cells; **d**, *p* = 2.26 × 10⁻²¹; *n* = 30 [continuous] and 15 [pulsed] cells; **e**, *p* = 1.71 × 10⁻⁷; *n* = 30 cells; **f**, *p* = 1.07 × 10⁻⁶; *n* = 30 cells and *n* = 10 liposomes). Insets in the right panels of (**c–f**) indicate areas under the curves (AUC) of normalized temperature relaxation. The unit a.u. in AUC means arbitrary units. ***P < 0.001 (Welch's two-sided t-test). Vertical scale bars in the right panels of (**c–f**) represent indicated temperature changes. ΔT was calculated by subtracting the average temperature from that of before heating. Source data are provided as a Source data file.

and Supplementary Movie 1). The comparison of the relaxation of averaged temperature after continuous heating of liposomes with that in an intracellular local space of the same size clearly showed that the temperature relaxation observed in single living cells was slower than that in water (Fig. 3f). Next, through several control experiments (Supplementary Note 4 and Supplementary Fig. 17), we confirmed the reliability of the FPT response (its insensitivity to factors other than temperature, such as viscosity) and the validity of IR laser irradiation regarding the aforementioned slow intracellular heat dissipation, ensuring that structural changes, distribution changes, or reactions—including physiological functions—of intracellular molecules do not induce artifacts. These results suggest that the slow temperature relaxation observed in cells is an intracellular component-dependent phenomenon.

Here, the time of temperature relaxation observed in cells after continuous heating (Fig. 3c–f) was compared with numerical calculations based on heat conduction. Numerical calculations based on the unsteady heat conduction equation, which consists of only a simple

thermal diffusion term, using the thermal conductivity and specific heat of cells, showed that the time required for the relaxation of the averaged temperature in cells by heat conduction is on the order of ms (Supplementary Fig. 18 and Supplementary Note 5). This comparison with the numerical calculation shows that the relaxation of the averaged temperature upon continuous heating observed in cells (which occurs on the order of seconds) is much slower than that by heat conduction.

## Conduction-independent, non-diffusive heat dissipation in living cells

Owing to the non-negligible difference in the relaxation time, we examined whether the intracellular temperature variation originates from heat conduction. Given the diffusive nature of heat, the temperature distribution immediately before the cessation of heating is essential for understanding the mechanism of temperature relaxation. For this purpose, we constructed a method to track the intracellular temperature distribution during heating and relaxation. Because the temperature mapping with high temporal resolution severely limits the number

of photons per pixel, we employed repeated measurements to achieve temperature distribution tracking in single cells (Fig. 4a, Supplementary Fig. 19, and Supplementary Movie 2), by taking advantage of the long-time measurement capability of this method (photobleaching-independence of fast lifetime) and the property of our thermal measurements where the temperature returned to its initial state.

Using this method, we investigated whether the characteristic dependency of the relaxation of intracellular averaged temperature on the heating durations shown in Fig. 3c, d, and Supplementary Fig. 11 are attributable to heat conduction. According to the heat conduction equation, where thermal properties remain constant, and there are no exothermic or endothermic sources, the rate at which the average temperature relaxes after heating stops is determined by the temperature distribution in the initial condition (Supplementary Note 6). Therefore, we investigated whether the average intracellular temperature relaxation rate, as observed in Fig. 3, depends on the initial temperature distribution. As shown in Fig. 4b, the average temperature relaxation rate was directly compared under the condition that the temperature distribution was kept constant immediately before heating was stopped (i.e., the initial condition), regardless of the heating time. This was achieved by adjusting the IR laser intensity. The results showed that the average temperature relaxation rate of the whole cell was slower with longer heating times, despite almost identical initial temperature distributions (Fig. 4c and Supplementary Movies 3–5). These results suggest that the rate of intracellular temperature relaxation is not determined by the initial temperature distribution; however, it is influenced by the heating duration, indicating that the intracellular temperature relaxation rate cannot be explained by heat conduction alone.

The results were then also verified with respect to the diffusive nature of heat transfer in heat conduction. If heat dissipation only occurs via conduction, the heat quantity decreases due to conduction from the region's boundary; therefore, for the same temperature distribution, the time constant for the average temperature relaxation rate depends on the size of the region. By using the equation (S3 and Supplementary Note 5) of heat conduction, we first numerically calculated and compared the average temperature relaxation rate of various areas ($D = 3$, 5, and 10 μm, where $D$ is the diameter). The calculated relaxation time of the average temperature in the finite region containing the heat source was found to increase with an increase in the region size (Fig. 4d and Supplementary Fig. 20). Next, we investigated this in living cells, examining the intracellular relaxation rate of the area-averaged temperature for observation areas of varying sizes centered on the heat source ($L = 3$, 5, and 10 μm, where $L$ is length of one side of the square area). In contrast to the numerical calculation based on heat conduction, the relaxation rate of the area-averaged temperature observed in the cells exhibited negligible dependence on the area size (Fig. 4e), exhibiting the distinctive characteristic of non-diffusive dissipation. The clear difference in the size dependence of the temperature relaxation rate between simulations based on the heat conduction equation and observations in living COS7 cells (Fig. 4f) indicates that intracellular heat dissipation does not depend solely on heat conduction. Therefore, it is suggested that, in addition to heat conduction, exothermic and endothermic processes contribute to the intracellular heat transfer pathway that governs in situ temperature change.

### Direct observation of high local temperature remaining in living cells during heat dissipation

Finally, to investigate the cause of the slow temperature relaxation in cells, we attempted to directly observe the slow relaxation of the temperature gradient in the intracellular microspace. Figure 5a, b shows the tracking of the temperature distribution of local areas when stopping IR laser irradiation. In both the cytoplasm (Fig. 5a and Supplementary Movie 6) and the nucleus (Fig. 5b and Supplementary Movie 7), the temperature gradient was maintained around the heating point for approximately 200 ms (cytoplasm) and over 300 ms

(nucleus) without immediate diffusion after the heat source disappeared. In addition to this, the temperature relaxation of the entire field of view was observed on the order of seconds. The temperature relaxation in the localized regions in the nucleus was slower than that in the cytoplasm (Fig. 5a, b), which is consistent with the findings on the tracking of cellular average temperature shown in Fig. 3. Furthermore, the difference in temperature relaxation in these two areas was clearly confirmed by the difference in the relaxation rate of the temperature gradient when heating was performed on the nuclear membrane (Supplementary Fig. 21). These results indicate that slow heat dissipation in the intracellular micro-space contributes to the slow temperature relaxation in cells.

## Discussion

In this study, we revealed that the measured average intracellular temperature slowly relaxes by using a high-speed temperature mapping method in cells through FPT and ultrasensitive FLIM, in contrast to the rapid relaxation of the average temperature in liposomes containing homogeneous aqueous solutions. Furthermore, we found that temperature relaxation depends on the intracellular location and its related molecules. Based on the temperature selectivity and fast response time of FPT, the results of control experiments using CP, the independence of the measurement methodology, and the fast relaxation at specific heating durations (e.g., 100 ms), we concluded that this significantly slower temperature relaxation measured inside cells compared to a homogeneous aqueous solution when cells were continuously heated is an intrinsic property of cells, and is not due to undesirable side effects of FPT. Additionally, by exploiting the high temporal and spatial resolution of temperature mapping, we observed transient changes in intracellular temperature distribution after stopping heating and discovered that slow and non-diffusive heat dissipation within cells does not depend on heat conduction. The slow intracellular temperature relaxation upon continuous heating was confirmed in the investigations in another mammalian cell line using HeLa cells (Supplementary Fig. 22).

By observing in situ heat transfer locally inside cells, we revealed that continuous heating induced a slow temperature relaxation and this temperature relaxation process does not follow the heat conduction equation with only a diffusion term (Fig. 4). Furthermore, based on the discussion in Supplementary Note 7 regarding the possibility that thermal resistance at multiphase interfaces within the cell slows heat dissipation, the slow thermal relaxation occurring on the order of seconds within the intracellular space cannot be explained by heat conduction alone, even when considering thermal resistance. This finding suggests that intracellular temperature changes are accompanied by the process of non-diffusive heat dissipation involving energy conversion in addition to heat conduction. Therefore, we reconsidered the energy dissipation process in cells during continuous heating. The average temperature relaxation rate in the intracellular region depends on intracellular organelles (such as the nucleus), RNA (Supplementary Fig. 14), and structures and large molecules that do not enter the GPMV (Supplementary Fig. 15). In addition, we observed that both heat dissipation from the inside to the outside of the cell and heat transfer in the intracellular space were slow (Fig. 5a, b). From these results, we speculate that the thermal energy generated by the absorption of IR laser light by intracellular water and biomolecules is propagated through energy absorption associated with state changes of specific molecules (related to the above structures) in the vicinity, in addition to being dissipated by heat conduction. This phenomenon may be attributable to the longer and thus non-negligible molecular relaxation times (compared with the relaxation times caused by heat conduction) of high-energy states of molecules (e.g., higher-order structures) induced by continuous heating. For instance, the intrinsic relaxation times of RNAs depend on their interactions[34]. When these relaxation times are longer than the relaxation time caused by heat

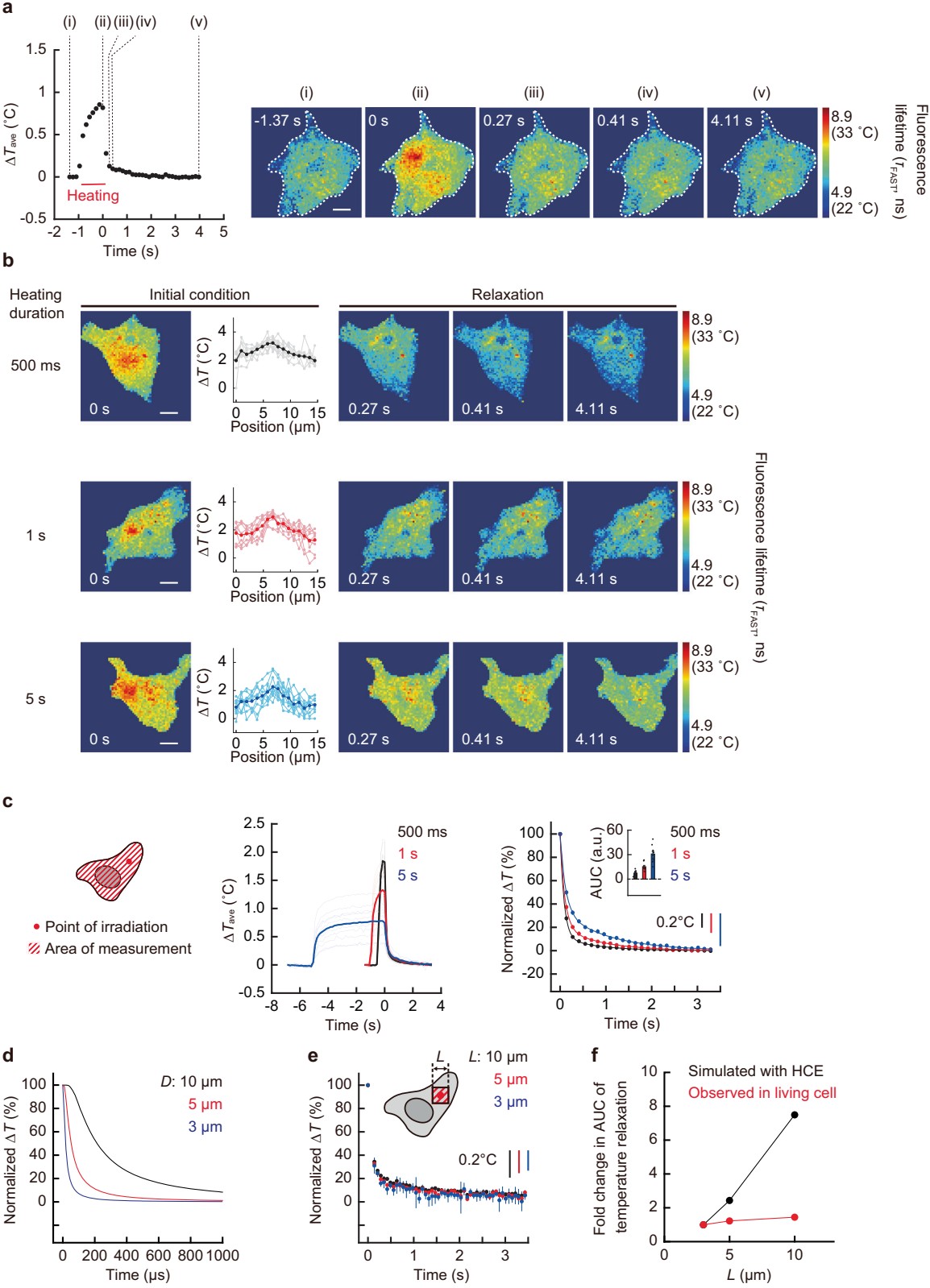

conduction (on the order of ms, see Supplementary Note 5), we can assume that the molecular relaxation of RNAs is a kinetic rate-limiting step in heat dissipation. As a result, this molecular relaxation dominates the heat dissipation of both RNAs and molecules that can share energy with RNAs. The thermal excitation of these biopolymers and the subsequent energy conversion keeps them in a higher energy state, preventing them from diffusive heat dissipation. However, considering

that measurements with different temperature probe molecules–which are expected to have distinct affinities to biomolecules, varying temperature-sensing chemistry, and different molecular relaxation times–recorded the same intracellular heat dissipation rates (Supplementary Figs. 12 and 13) and intracellular temperature changes on the same order[1–16,18–20], it is unlikely that molecular relaxation of the probe molecules affects the measurement of intracellular temperature. As

**Fig. 4 | Independence of intracellular temperature relaxation after continuous heating from the heat conduction model. a** The temperature mapping of a whole cell with local heating (0.55 mW for 1 s). The tracking of average temperature change (left) and representative temperature distributions in cells at various time points. The cell is indicated by a dotted line (right). Scale bar represents 10 μm. The experiment was performed four times with similar results. **b** The intracellular temperature distribution with different heating conditions (top [0.89 mW for 500 ms], middle [0.83 mW for 1 s], and bottom [0.55 mW for 5 s]). The line profiles of $\Delta T$ of cells and the temperature mapping during relaxation are also shown. Scale bars represent 10 μm. The experiment was performed two (top), four (middle), and five (bottom) times, respectively, with similar results. **c** The tracking of average temperature change of the whole cell, normalized temperature relaxation, and its area under the curve (AUC, inset). The unit a.u. in AUC means arbitrary units. Data in

the inset are mean ± s.e.m. ($n$ = 10 [500 ms], 9 [1 s], and 10 [5 s] cells). Vertical scale bars represent temperature change. **d** The average temperature relaxation for various spatial sizes was analyzed from the time course of the temperature distribution numerically calculated by the heat conduction equation (HCE) in Supplementary Note 5. Areas of diameter ($D$) = 3, 5, and 10 μm are shown. **e** The relaxation of average temperature of various-sized regions observed in the cytoplasm of living COS7 cells after continuous heating (5 s) (Fig. 3d). Data in cells are presented as mean ± s.e.m. ($n$ = 30 cells). Areas of a square with a side ($L$) = 3, 5, and 10 μm are shown. **f** Comparison of the size dependence of temperature relaxation times evaluated using fold change in area under the curve (AUC) between the simulation based on the HCE and the observation in living COS7 cells. **a**–**c** $\Delta T$ was calculated by subtracting the temperature from that of after the temperature relaxation. Source data are provided as a Source data file.

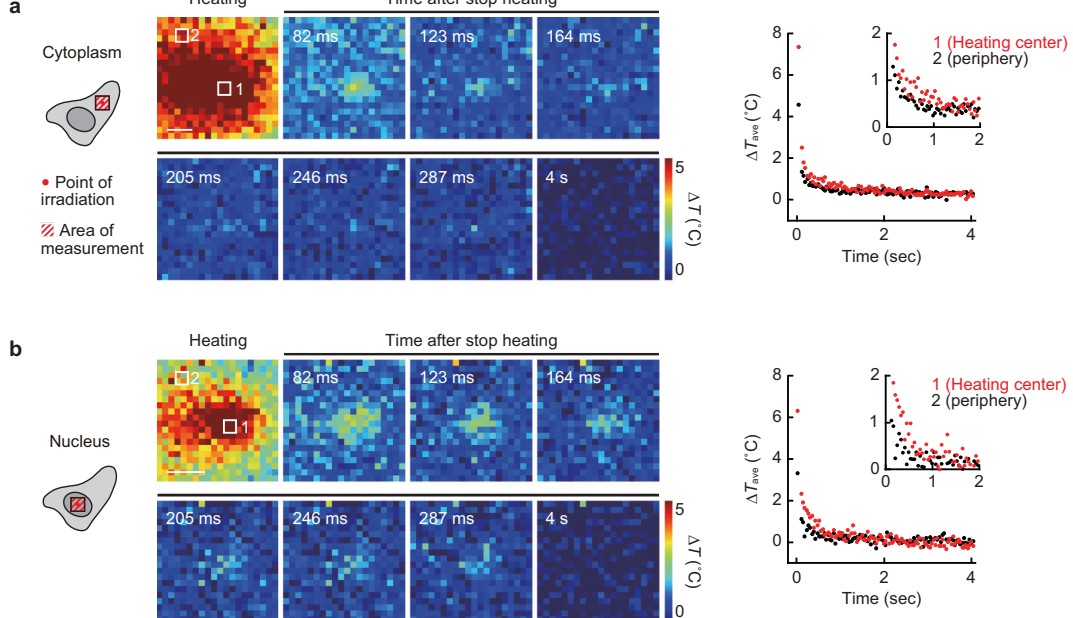

**Fig. 5 | Observation of slow temperature relaxation in the intracellular local area.** High-speed mapping of intracellular temperature in the cytoplasm (**a**) and in the nucleus (**b**) immediately after stopping local heating (1.1 mW [cytoplasm] and 0.83 mW [nucleus] for 5 s). Fluorescence lifetime images were taken every 41 ms. The right graphs show the relaxation of average temperature in the

heating areas indicated in the left images. $\Delta T$ was calculated by subtracting the temperature from that of after the temperature relaxation. Scale bars represent 2 μm. The experiments were performed three times with similar results. Source data are provided as a Source data file.

similar temperature measurements can be made regardless of the type of thermometer, the implication is that energy is shared among biopolymers, each of which possesses unique dynamics.

On the other hand, an equilibrium system also exists within the cell in which heat is dissipated by conduction. In fact, our pulsed heating experiments and the recently developed label-free temperature measurement by refractive index change showed a fast temperature relaxation and a thermal conductivity not much different from that of water[27]. These facts suggest that intracellular temperature change includes both the temperature defined by a group of biomolecules under nonequilibrium conditions and the temperature governed by the heat-conduction equation, which relies solely on thermal diffusivity. Examples of such nonequilibrium relationships between systems with different mechanisms of heat transfer (i.e., different temperature-determining populations) can be found in plasmas (electrons and cations)[35] and metal crystals (free electrons and phonons)[36]. In these examples, as the rate of energy conversion between systems in nonequilibrium states is not determined by the thermal conductivity of each system, it is possible to define different temperatures for the two systems within a time shorter than the relaxation time of that energy conversion. In cells, because of the

energy conversion of certain biomolecules through state changes and their subsequent relaxation, resulting in low heat transfer rates, it is possible to define different temperatures for the two systems within the time required for energy dissipation. The elucidation of the specific mechanisms of energy conversion and molecular relaxation of biomolecules that generate this hypothetical temperature difference between the two systems may lead to an accurate definition of intracellular temperature, encompassing nonequilibrium conditions—an area that requires further exploration in future thermodynamic investigations.

To understand heat transfer in biological samples of various sizes, such as cells, tissues, and organs, it is necessary to consider the spatial scale of the observations. The time required for the temperature variations within the entire system to disappear via heat conduction strongly depends on the size of the space. On the other hand, as biological macromolecules have unique conformational change timescales that depend on their structure[34], the relaxation times of thermally excited molecules are independent of the size of the space. Therefore, the time required for heat dissipation via heat conduction including the effects of interfacial thermal resistance in spaces of certain sizes (i.e., in large spaces) is non-negligible compared with the

relaxation time of the molecules. The relaxation time of subcutaneous tissue temperature observed in mice using a luminescent nano-thermometer showed little deviation from the heat conduction model[37]. The phenomenon revealed in this study, i.e., that the intracellular temperature change is not completely governed by heat conduction, is observed only in the single-cell space. This suggests that, especially in cells, it is inappropriate to assume that all thermal energy is dissipated via heat conduction when determining the thermal conductivity and other thermal properties from temperature changes, given the substantial effects of energy conversion and molecular relaxation of biomolecules. The cellular thermal conductivity recently estimated from the local temperature change of molecular thermometers[5,23,24] differed significantly from the cellular thermal conductivity determined from a measurement in bulk (cell suspension) at a relatively large spatial scale[25] and equilibrium thermal measurements[26,27]. The low thermal conductivities and the thermal diffusivity estimated in cells using the former methods should reflect the significant influence of molecular relaxation on temperature changes at the single-cell scale, as discovered in this study.

Until now, the physical mechanism causing the discrepancy (the $10^5$ gap) between the temperature change inside cells and the temperature change calculated by the heat conduction equation of a homogeneous system was unknown. The discovery of the existence of a very slow heat dissipation process that is independent of conduction highlights a critical flaw in attempts to estimate the temperature change during spontaneous intracellular heat generation from thermal conductivity and the heat conduction equation. In particular, regarding the large discrepancy of the $10^5$ gap issue, our findings challenge the basic assumption that intracellular temperature changes are caused solely by simple heat conduction. Slow intracellular non-diffusive heat dissipation discovered in this study (Fig. 4), which cannot be described by the simple heat conduction equation, can explain the temperature change during intracellular heat generation that lasts for more than 1 s. In fact, cells with slower heat dissipation had a greater temperature increase during endogenous heat generation (i.e., mitochondrial stimulation) (Supplementary Fig. 23). In addition, we estimate an intracellular temperature increase of 1.1 °C when heat dissipates independent of conduction (i.e., the cell can be assumed to be adiabatic homogeneous during the relaxation time) upon mitochondrial stimulation (Supplementary Note 8). This temperature increase is in excellent agreement with the intracellular temperature increase measured in both the present (Supplementary Fig. 23) and previous studies[15,16]. This consistency demonstrates the importance of considering the thermal excitation of intracellular biopolymers and their energy dissipation processes, rather than relying solely on simple heat conduction equations when attempting to understand the mechanisms of intracellular temperature change.

Furthermore, the temperature elevation of biomolecules (which can be sensed by molecular thermometers) induced via slow intracellular heat dissipation has physiological significance. The slow heat transfer during continuous artificial heating observed in this study is also expected to occur during relatively long-term (on the order of minutes) physiological heating within cells, such as mitochondrial heating (Supplementary Fig. 23). Our study has reported that spontaneous heat generation, which was dependent on the activity of neurons, caused an intracellular temperature rise ($\Delta T = 2.12$ °C), activating its own thermosensitive transient receptor potential vanilloid 4 (TRPV4)[38]. More recently, a thermal signaling mechanism has been discovered in neuronal cells where spontaneous temperature increase ($\Delta T = 0.9$ °C) in the nucleus has a positive effect on important developmental processes during neuronal differentiation[9]. The slow intracellular non-diffusive heat dissipation could explain how heat generated at the single-cell level contributes to these thermal signaling mechanisms, which are activated by the aforementioned high

temperatures. Thus, slow heat dissipation in cells may play a pivotal role in facilitating the temperature rises that trigger various physiological cellular functions[9,18–20]. This new concept should help to elucidate the intracellular signaling mechanisms involved in dynamic medical phenomena (e.g., epilepsy[38,39], cancer hyperthermia[40], and inflammation[41]) and cell biological events (body temperature control by brown adipose tissue[15,16], embryogenesis[42,43], neuronal activity[9,19,44], apoptosis[1], and liquid–liquid phase separation[45]) that are closely related to temperature changes. Through these investigations, the concepts discovered in this study should provide a revolutionary perspective on life science research that has long been solely based on molecular communication.

# Methods

## Cell culture
COS7 cells (no. RCB0539, RIKEN BRC, Tsukuba, Japan) and HeLa cells (no. RCB0007, RIKEN BRC) were cultured in Dulbecco's Modified Eagle's Medium (DMEM) with 10% fetal bovine serum (FBS) supplemented with penicillin–streptomycin, L-glutamine, sodium pyruvate, and nonessential amino acids at 37 °C in 5% $CO_2$. For live-cell imaging, cells were cultured in 35-mm glass-bottomed dishes (AGC Techno Glass, Shizuoka, Japan), and the medium was replaced by phenol red-free culture medium containing HEPES buffer (2 mL) before live-cell imaging. All solutions were from Thermo Fisher Scientific (MA, USA).

## Microinjection into living cells and culturing on the microscope
Microinjection of FPT, CP, Rhodamine B-labeled dextran (molecular weight 10,000; Thermo Fisher Scientific), and Cy3-tagged streptavidin (FUJIFILM Wako Pure Chemical Corporation, Osaka, Japan) into living cells was performed with Femtojet (Eppendorf, Hamburg, Germany), controlled by a micromanipulator (Eppendorf). FPT (0.5 w/v%), CP (0.5 w/v%), Rhodamine B-labeled dextran (20 μM), and Cy3-tagged streptavidin (3 μM) were dissolved in an aqueous solution containing 140 mM KCl, 10 mM $KH_2PO_4$–$K_2HPO_4$, and 4 mM NaCl. The solution was filtered using an Ultrafree-MC filter (Merck Millipore, MA, USA) and microinjected into the cytoplasm with a glass capillary needle (Femtotips II; Eppendorf). The volume of injected solution was estimated to be 2 fL.

The temperature of the culture medium was maintained at 26 °C for monitoring temperature change and at 24 °C for imaging temperature distribution. We chose these temperatures because they are the temperatures associated with the highest sensitivity for measuring temperature with FPT (see Fig. 1c) in each experiment. The temperature of the culture medium was controlled and monitored with a stage-top incubator and a microscope objective lens heater (INUBSF-ZILCS; Tokai Hit, Fujinomiya, Japan).

## Preparation of cell extract
The COS7 cell pellets were collected and resuspended in a buffer (30 mM HEPES (pH 7.4), 75 mM sucrose, 225 mM mannitol, and 0.1 mM EGTA). After the addition of protease inhibitor cocktail (EDTA-free, Nacalai Tesque, Inc, Kyoto, Japan), the cell suspend was lysed using Dounce Tissue Grinder (Wheaton, Millville, NJ), followed by three times of centrifugation (600 × $g$, 10 min, 4 °C). Following further centrifugation (10,000 × $g$, 10 min, 4 °C) of the supernatant, KCl was added to make the ionic strength 0.3 M. The concentration of proteins in the obtained cell extract was quantified with the BCA Protein Assay Kit (Thermo Fisher Scientific) after treated with 1% Triton X-100 (Sigma-Aldrich, St. Louis, MO) to be 2.9 g $L^{-1}$.

## Fluorescence lifetime imaging of living cells, cell extracts, and GPMVs
TCSPC-based fluorescence lifetime imaging microscopy (FLIM) was performed on a TCS SP8-FALCON confocal laser-scanning microscope

(Leica Microsystems) with a pulsed diode laser (PDL 800-B, 470 nm; PicoQuant, Berlin, Germany) at a repetition rate of 20 MHz, and the emission from 500 to 640 nm was obtained through the described objective with 1–18.2 zoom factors and a binning procedure (factor: 1 or 2) in a $64 \times 64$–$1024 \times 1024$-pixel format for 0.7–257.5 s. To obtain the conventional fluorescence lifetime ($\tau_f$), the fluorescence decay curves of FPT in single cells and cell extract were fitted with a double exponential function:

$$I(t) = A_1 \exp(-t/\tau_1) + A_2 \exp(-t/\tau_2) \quad (1)$$

Then, the conventional fluorescence lifetime ($\tau_f$) was calculated using following equation:

$$\tau_f = (A_1\tau_1^2 + A_2\tau_2^2)/(A_1\tau_1 + A_2\tau_2) \quad (2)$$

In determining the intracellular temperature change ($\Delta T$) during heating by an artificial heat source, we used the temperature response curve of $\tau_{FAST}$ of FPT (Supplementary Fig. 1) for calibration when the entire cell was heated to ignore the temperature increase due to the steady-state spontaneous heat generation of cells. For intracellular temperature mapping, we used a calibration using cell extracts[2] (Fig. 1c). The temperature response curves for the temperature mapping with FPT in single cells and cell extract were obtained by approximating the relationship between the averaged fast fluorescence lifetime ($\tau_{FAST}$) of FPT (in triplicate) and the temperature using fourth- to sixth-degree polynomials (coefficients of determination for both $R^2 = 0.99$):

$$T = A_1\tau_{FAST}^6 + A_2\tau_{FAST}^5 + A_3\tau_{FAST}^4 + A_4\tau_{FAST}^3 + A_5\tau_{FAST}^2 + A_6\tau_{FAST} + A_7 \quad (3)$$

where $T$ and $\tau_{FAST}$ represent the temperature (°C) and the fast fluorescence lifetime (ns), respectively. The coefficients ($A_1$–$A_7$) in Eq. 3 are listed in Supplementary Table 1.

### Fluorescence imaging of the cells

Confocal fluorescence imaging was performed on a TCS SP8-FALCON confocal laser-scanning microscope (Leica Microsystems, Wetzler, Germany) with a laser (638 nm) and an HC PL APO 63×/1.40 Oil CS2 objective (Leica Microsystems). Mitochondria and the nucleus were labeled with MitoTracker Deep Red FM (Thermo Fisher Scientific) and SiR-DNA (Cytoskeleton, Inc., CO, USA), respectively, and the fluorescence from 650 to 750 nm was obtained.

Confocal fluorescence imaging of Rhodamine B and Cy3 were performed on a TCS SP8-FALCON confocal laser-scanning microscope (Leica Microsystems) with a laser (552 and 488 nm), and the fluorescence from 560 to 700 nm and 500 to 700 nm were obtained, respectively.

### Monitoring transient heating with IR laser irradiation

An IR laser (FRL-DC 1480 nm, maximum power, 3 W; IRE-Polus, MA, USA) was installed in a TCS SP8-FALCON confocal laser-scanning microscope (Leica Microsystems) through the IR-LEGO system (Sigma Koki, Tokyo, Japan) for the local heating of living cells and liposome, as shown in Fig. 2a. An HC PL APO 63× 1.40 N.A. OIL CS2 objective (Leica Microsystems) was used to focus the IR laser and image the fluorescence lifetime of specimens; the diameter of the focused area in the IR laser irradiation was estimated to be 1.29 μm. An electronically controlled shutter (SSH-C4B; Sigma Koki) was used to start and stop the irradiation of the IR laser. The durations of irradiation were 0.1–5 s. The timing of stopping heating was analytically determined by the rapid drop in FPT fluorescence intensity.

Considering that the heating operation using the shutter and the timing of the scanning in imaging are not perfectly synchronized, the first frame after the fluorescence lifetime drop was excluded because it may not reflect the temperature.

### Absorption spectra of FPT

Absorption spectra of FPT (0.1 w/v% in KCl aqueous solution [150 mM]) were obtained using an IRT-5200 infrared microscope (JASCO, Tokyo, Japan). A sample fixture was installed in a handmade cryostage for the microscope, and the sample temperature was stably controlled at 25 °C using liquid nitrogen and an electric heater integrated into the cryostage[46].

### Tracking structural transition of FPT

The response time ($\tau$) of the thermometers was determined by approximating with a single exponential function:

$$F(t) = a \exp(-t/\tau) + b \quad (4)$$

where $F(t)$ and $t$ represent the fluorescence intensity and time (s), respectively, and $a$ and $b$ are constants.

### Fluorescence recovery after photobleaching (FRAP) analysis of FPT in living COS7 cells

The mobility of FPT in living cells was investigated by fluorescence recovery after photobleaching (FRAP). In the FRAP experiments, a small round area in living COS7 cells microinjected with FPT was photobleached for 2 s with 488 nm laser light focused on the area in the cytoplasm, followed by time-lapse confocal fluorescence imaging (described in "Methods") over 200 s. In the image analysis, the fluorescence recovery was determined using the ratio of the averaged fluorescence intensity of the photobleached spot area to that of the whole cytoplasm, calculated for each time point. The time course of fluorescence recovery for FRAP employing a uniform circular beam was fitted using the formula of Soumpasis et al.[47].

### Inhibition of endogenous metabolic thermogenesis

Endogenous thermogenesis via glycolysis and oxidative phosphorylation was inhibited by adding 10 mM 2-deoxy-D-glucose (Sigma-Aldrich), 1 μM rotenone, and 1 μM antimycin A (Sigma-Aldrich) to the cell observation medium immediately before the experiment.

### Preparation of giant plasma membrane vesicles (GPMVs)

COS7 cells injected with FPT were washed twice, then incubated with PFA (25 mM) and DTT (2 mM) dissolved in buffer (10 mM HEPES [pH 7.4], 150 mM NaCl, and 2 mM CaCl$_2$) for 3 h at room temperature.

### Preparation and fluorescence imaging of liposome

Briefly, 1,2-dioleoyl-sn-glycero-3-phosphocholine (DOPC) was dissolved in chloroform/methanol mixtures (2/1, v/v). Then, 20 μL of DOPC solution (5 mM) was poured into a glass test tube and gently dried using a flow of air to produce a thin DOPC film at the bottom of the glass tube. This DOPC film was subsequently dried under a vacuum for 2 h. It was then hydrated with 190 μL of Rhodamine B-labeled dextran (10 μM) in a 200 mM sucrose solution at 50 °C overnight after 5 min of prehydration (50 °C) using 10 μL of the solution. The final DOPC concentration was 0.5 mM.

To prevent liposomes from moving out of the field of view upon IR laser irradiation due to convection, a micro-sized chamber array[48] of 20 μm in diameter and 50 μm in depth was introduced, and a single liposome was placed in the well. First, a cover slip, on which microchambers were fabricated with SU-8, was treated with 200 μL of 200 mM glucose solution and deaeration, followed by

the application of 2 µL of the solution containing liposomes and 18 µL of 200 mM glucose solution. Then, this cover slip with microchambers was sealed with a cover glass to allow fluorescence lifetime imaging.

Confocal fluorescence imaging of liposomes was performed on a TCS SP8-FALCON confocal laser scanning microscope (Leica Microsystems), as described above. The fluorescence signal observed inside the liposome using confocal microscopy confirmed the inclusion of Rhodamine B-labeled dextran. Temperature changes were determined from the temperature-dependent fluorescence intensity of Rhodamine B in aqueous solution (Fluorescence intensity $= 1.8294 e^{-0.0229\,T}$, where $T$ indicates the temperature; $R^2 = 0.99$).

### Comparison of the time-course of temperature relaxation
The time-course of the temperature relaxation normalized by the temperature increase just before cessation of the heating by IR laser irradiation was analyzed. The line of fitting with an exponential function is shown for reference. The area under the curve (AUC) of normalized temperature relaxation for 3.5 s was also calculated, followed by statistical analysis. The first two points were excluded in the calculation of AUC because the time required for the structural relaxation of FPT does not reflect the temperature (Supplementary Fig. 8).

### Treatment with Actinomycin D
The chemical stimulus was conducted by adding 2 mg L$^{-1}$ Actinomycin D (Nacalai Tesque, Inc., Kyoto, Japan) to the culture medium 6 h prior to the observation.

### Tracking of intracellular temperature distribution upon IR laser heating
Because the temperature mapping with high temporal resolution severely limits the number of photons per pixel, we tracked the intracellular temperature distribution by repeating the measurements on the same cells (Supplementary Fig. 19). Fluorescence lifetime movies lasting 55 s (heating duration: 500 ms) and 105 s (heating duration: 1 and 5 s), including the duration of IR laser irradiation, were recorded five times in the same cells (Fig. 4a, b) and once in 10 cells (Fig. 5a, b) or 20 cells (Supplementary Fig. 22d), and averaged at the timing when IR laser irradiation was stopped.

### Measurement of temperature change during mitochondrial thermogenesis
The mitochondrial stimulus was conducted by quickly adding 1 mL of 200 µM FCCP (Sigma-Aldrich) in FBS-free culture medium to the studied cells in 1 mL of FBS-free culture medium in a glass base dish. The temperature of the culture medium was maintained at 26.0 °C.

### Statistical analysis and reproducibility
Statistical significance was determined by Welch's two-sided t-test. No statistical method was used to predetermine the sample sizes. The experiments were not randomized. The investigators were not blinded to allocation during experiments and outcome assessment. Cells displaying significant morphological alterations following microinjection, laser heating, or treatment with inhibitors or cells in which the signal (fluorescence intensity or lifetime) from the fluorescent probes did not return to preheating levels, were excluded from the analysis. Data indicating fluorescence lifetime values outside the range of the temperature response curve with FPT were excluded.

### Reporting summary
Further information on research design is available in the Nature Portfolio Reporting Summary linked to this article.

## Data availability
All data that support the findings of this study are included in the paper. The raw data acquired by FLIM exceeds 200 GB and cannot be deposited in a public repository due to size limitations. All data are preserved by the authors and can be accessed by contacting the corresponding author. Requests will receive a response within 1 month. Source data are provided with this paper.

## Code availability
The scripts used in this study are available online at: https://github.com/tkrdmshr/Non-diffusive_slow_heat_dissipation_induces_high_local_temperature_in_living_cells.git. The scripts are also available at https://doi.org/10.5281/zenodo.18934454.

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

## Acknowledgements

We thank Professor Katsunori Hanamura, Professor Takeharu Nagai, Professor Takuro Ideguchi, and Dr. Keiichiro Toda for critical comments on this manuscript. We are also grateful for financial support from PRESTO and CREST of JST, JSPS KAKENHI (18H03981, 20H05785, 21J14440, 24H02306, and 25K02236), Life Science Foundation of Japan, and Brain Science Foundation.

## Author contributions

K.O. organized this work. K.O., M. Takarada, M. Takinoue, and T.F. designed this work. M. Takarada, K.O., performed all imaging experiments and analyzed the data. M. Takarada, M. Takinoue, M.I., and M.M. prepared and performed experiments using liposomes. K.V.T., H.N. designed and fabricated the microarray chambers. R.S. obtained absorption spectra, performed numerical calculations, and contributed to the interpretation of the results. K.O., M. Takarada wrote the paper.

## Competing interests

The authors declare no competing interests.

## Additional information

**Supplementary information** The online version contains Supplementary material available at https://doi.org/10.1038/s41467-026-71878-y.

