## [Transparent Peer Review File · Nature Communications]

Non-diffusive slow heat dissipation induces high local temperature in living cells

Corresponding Author: Professor Kohki Okabe

Version 1:

Reviewer comments:

Reviewer #1

(Remarks to the Author)

By high-speed intracellular temperature mapping, this work claims that cells exhibit significantly slower temperature relaxation after continuous heating compared to liposomes, and identifies the existence of intracellular energy dissipation independent of heat conduction. However, this work has some important logical concerns and lacks innovation.

1. In this paper, the authors used the FPT fluorescent probe to detect intracellular temperature. The authors claim that the probe is highly sensitive to temperature, but there are numerous factors influencing the probe's lifetime, such as burial degree, diffusion rate of the probe, collision probability with other molecules, etc., all of which can significantly affect the fluorescence lifetime of the probe. How did the authors eliminate the influence of other factors? Especially in the complex environment of cells, how to correlate fluorescence lifetime with temperature as a single factor is the primary issue to be addressed.
2. In Figure 1c, the authors compared the relationships between the fluorescence lifetimes of FPT in cells calculated by different methods (conventional method and fast method) and the temperature of the cellular medium. This is confusing because the authors aim to detect intracellular temperature through the probe's lifetime—that is, intracellular temperature is supposed to be calculated from the probe's lifetime. What is the purpose of analyzing their correlation?
3. The authors' method for verifying the fast lifetime calculation (Figure 1d) is also highly confusing. Since the traditional method requires exponential function fitting, it needs more photons to obtain a convergent result, so the standard deviation (SD) must be large when the number of photons is small. In contrast, the authors calculate errors using an average value in a linear manner, resulting in a smaller SD. Therefore, the conclusion "As shown in Figure 1d, the determination of the fluorescence lifetime of FPT by fast lifetime achieved similar accuracy with only 4% of photons compared to the conventional method" is too arbitrary.
4. In this work, the authors irradiated living COS7 cells with an IR laser and studied intracellular temperature. How can they ensure that the laser affects only temperature rather than the structural characteristics or physiological functions of living cells, such as the enrichment of substances in intracellular regions? Which might also contribute to the fluorescent probes.
5. In addition to the doubtful conclusions, the paper's innovativeness is also limited. Merely changing the calculation method for fluorescence lifetime lacks theoretical basis or innovation and fails to provide new mechanistic solutions.

Besides, some minor concerns :

1. The language in the paper is not well organized and optimized. For example: "FLIM using the most accurate and pervasive TCSPC was used for detecting the fluorescence lifetime of intracellular FPT."
2. The formatting of formulas in the methods section needs revision, and the formulas and the parameters could be separated.
3. The Introduction section is overly simplistic, failing to fully elaborate on the background and current research progress. Some key literature is also not cited.

Reviewer #2

(Remarks to the Author)

Takarada et al used fast FLIM imaging method to measure the dynamics heat dissipation inside cells. It was found that the heat dissipation inside cells is significantly longer than the time scale predicted by heat conduction theory. In terms of methodology, the fast FLIM method they used significantly improved the temporal resolution of fluorescent temperature

measurement, enabling the tracking of the heat dissipation process inside cells on ms time scale, which is very impressive. From a scientific discovery perspective, the observation of such a long time-scale heat decay inside cells is surprising and exciting. This may imply that the thermodynamics inside cells do not conform to traditional theoretical models, if the experimental measurements are credible. This will undoubtedly update our understanding of the thermodynamics of living systems. Overall, I strongly recommend the publication of this work. To enhance the strength of this work, I have the following suggestions:

1. The author's team is one of the pioneers in the use of fluorescence thermometry to measure cell temperature and has carried out a series of exploratory work on cell thermodynamics. The FPT they developed is also a very typical fluorescence thermometer. I appreciate these contributions. Recently, the measurement uncertainty of fluorescence thermometers has been noticed. Especially for small-molecule fluorescence thermometers, how to ensure that the measured spectral changes (such as lifetime) depend solely on temperature but not other factors? This seems difficult because, from the principles of photo-physical chemistry, in addition to temperature, the viscosity of the medium is also an important factor in fluorescence lifetime. Can the authors comment on this issue? For example, is the calibration curve shown in Figure 1 carried out in solution or in cells? How they fix this issue on their FPT?
2. Are there any control experiments in the medium to see how long the heat dissipation time scale is in known media and whether it is related to the thermal conductivity and thermal diffusion of the medium? These control experiments are very important. On the one hand, it can argue for the effectiveness of the method, and on the other hand, it can also compare with the situation inside cells. I strongly suggest the authors to try heat dissipation tracking experiments in water and glycerol at different proportions to see the time scale of heat dissipation, and how the timescale related to their heat conductivity? There will definitely be interesting results.
3. I have a technical concern about fast FLIM. Apparently, to increase the time resolution, the accumulation of photons is reduced, as shown in the methods section of the paper. With the decrease in photons, shot noise will increase significantly, which in turn will increase the error in lifetime fitting. Has the author quantitatively assessed the error range of fast FLIM, in principle and in theory, and is it within an acceptable level? Of course, the present data shows some comparisons of individual curves and they fit pretty wells, but that is not enough to convince me completely.
4. In addition to fluorescence thermometers, there are also other methods to try to measure the heat dissipation and thermal relaxation inside cells. For example, in the (Song et al. Chem, 2021, 7-6, 1569-1587) literature, a heated nanoprobe was used to observe a heat relaxation process in cells, and they also observed significant slower thermal decay than the timescale predicted by the heat conduction model (>two orders of magnitude). Although the methods differ, the main results seem point in the same direction. I suggest that the authors cite this literature and briefly discuss this point.
5. Regarding why there is such slow heat relaxation inside cells, in addition to the energy conversion possibilities discussed by the authors, I have some different views. In fact, the classical laws of heat transfer, including Fourier's law of heat conduction and Newton's law of cooling, were derived and verified in macroscopic, homogeneous media. For micro-nano scales with complex multiphase compositions, such as cells, the above theoretical models may show significant deviations, or even be completely inapplicable. In the case of cells, the complex organelles and locally concentrated biomacromolecules (phase separation) inside cells create complex multiphase interfaces. These interfaces create interfacial thermal resistance (Kapitza thermal resistance) for heat transfer. Perhaps the Kapitza thermal resistance on one interface is negligible, but there are thousands of such interfacial thermal resistances inside cells, which will significantly slow down the dissipation of heat. I suggest that the authors also consider this possibility, if the length of the discussion allows.
6. Some detailed errors, the content of Figure 4b seems inconsistent with the corresponding figure legend description (top, middle, bottom); There is an absence of a space between numerical values and the degree Celsius symbol; in certain expressions, such as "2.9 mgmL⁻¹", which clearly requires standardization.

Version 2:

Reviewer comments:

Reviewer #1

(Remarks to the Author)

All my concerns have been addressed. I appreciate the responses of the authors.

Reviewer #3

(Remarks to the Author)

In my first-round review, I raised many challenging questions; the authors have addressed them with thorough revisions and new experimental evidence, and they have, on the whole, satisfactorily clarified my concerns. Although the precise origin of the remarkably slow heat dissipation inside cells remains incompletely understood, the phenomenon reported in this work is truly surprising. I believe these findings will significantly deepen our understanding of the intracellular thermodynamic environment. I therefore maintain my initial recommendation: I strongly support publication.

[Point-by-point responses and revisions in accordance with reviewer's comments]

Ms ID: NCOMMS-24-79782B

Ms title: Non-Diffusive Slow Heat Dissipation Induces High Local Temperature in Living Cells

Reviewer #1 (Remarks to the Author)

Comment: By high-speed intracellular temperature mapping, this work claims that cells exhibit significantly slower temperature relaxation after continuous heating compared to liposomes, and identifies the existence of intracellular energy dissipation independent of heat conduction. However, this work has some important logical concerns and lacks innovation.

Response: We appreciate that Reviewer #1 properly summarized our manuscript and gave constructive ideas for the revision.

Comment: In this paper, the authors used the FPT fluorescent probe to detect intracellular temperature. The authors claim that the probe is highly sensitive to temperature, but there are numerous factors influencing the probe's lifetime, such as burial degree, diffusion rate of the probe, collision probability with other molecules, etc., all of which can significantly affect the fluorescence lifetime of the probe. How did the authors eliminate the influence of other factors? Especially in the complex environment of cells, how to correlate fluorescence lifetime with temperature as a single factor is the primary issue to be addressed.

Response: We deeply appreciate your insightful comments on our research. When developing FPT, we prioritized ensuring that its fluorescence lifetime response was derived solely from temperature and unaffected by chemical, physical, or biological factors. In our previous work, we demonstrated that FPT selectively detects intracellular temperature changes due to its functional independence; FPT did not respond to physicochemical factors other than temperature, such as pH, ionic strength, biomolecular concentration, and viscosity, when they were quantitatively altered in solution (Ref. 2 in the revised manuscript). Furthermore, we used a control copolymer (CP) with a chemical composition almost identical to that of FPT, but lacking temperature sensitivity, to confirm the temperature selectivity of FPT. In complex environments like the intracellular space, various physicochemical changes can occur due to heating. Therefore, rigorous verification using this CP is essential to achieving our objectives. The inclusion of a control probe that enables the investigation of temperature selectivity is a unique feature of FPT that is not found in other nanothermometers. Thus, we added an explanation of this functional independence of FPT to the manuscript and investigated the response of CP during intracellular heating. Through various investigations using CP, we demonstrated that FPT is unaffected by changes in the diffusion rate of intracellular FPT or the probability of collisions with other molecules, which may vary during heating. This strong evidence that the intracellular FPT response is due solely to temperature changes strengthens our claim in this paper; thus, we added these results to the revised manuscript.

Texts and figures added to the revised manuscript (Results, Discussion sections, and Supplementary Note 4):

“We previously demonstrated that FPT selectively detects intracellular temperature changes due to its functional independence; FPT did not respond to physicochemical factors other than temperature, such as pH, ionic strength, biomolecular concentration, and viscosity, when they were quantitatively altered in solution². In this study, we used a control copolymer (CP)² with a chemical composition almost identical to that of FPT, but lacking temperature sensitivity, to confirm the temperature selectivity of FPT. In complex environments like the intracellular space, various physicochemical changes can occur due to heating. Therefore, rigorous verification using this CP is essential to achieving our objectives. The inclusion of a control probe that enables the investigation of temperature selectivity is a unique feature of FPT that is not found in other nanothermometers.” (Line 93-101 in the revised manuscript)

“We confirmed that CP does not exhibit temperature dependence even in fast lifetime measurements (Figure 1d).” (Line 113-114 in the revised manuscript)

“Furthermore, the analysis of the fluorescence decay curve of CP confirmed that factors other than temperature (e.g., viscosity) induced by IR laser irradiation do not directly affect the FPT response (Supplementary Figure 5).” (Line 149-151 in the revised manuscript)

“On the other hand, IR laser irradiation did not change the fluorescence lifetime distribution of CP (Figure 2c). These results clearly demonstrate that the spatial distribution of the fluorescence lifetime of FPT created by IR laser irradiation is solely due to heat, independent of the chemical environment.” (Line 153-156 in the revised manuscript)

“Using CP, we confirmed that FPT responds only to temperature changes induced by IR laser irradiation (Figure 3b).” (Line 183-184 in the revised manuscript)

“Based on the temperature selectivity and fast response time of FPT, the results of control experiments using CP, the independence of the measurement methodology, and the fast relaxation at specific heating durations (e.g., 100 ms), we concluded that this significantly slower temperature relaxation measured inside cells compared to a homogeneous aqueous solution when cells were continuously heated is an intrinsic property of cells, and is not due to undesirable side effects of FPT.” (Line 298-302 in the revised manuscript)

“(a) Temperature selectivity of the FPT response during IR laser irradiation

During heating, factors other than temperature, such as viscosity, also change. Previously, we confirmed that the fluorescence response of FPT is temperature-selective and remains unaffected by factors such as pH, ionic strength, biomolecule concentration, and viscosity². In this study, we investigated the temperature selectivity of the FPT response during IR laser irradiation

First, because FPT does not directly absorb IR laser light (Supplementary Figure 3), its response is independent of changes due to electronic excitation. Furthermore, CP having the same chemical composition as FPT but lacking temperature responsiveness did not respond to IR laser irradiation (Figures 2c, 3b, and Supplementary Figure 5). These results rule out the possibility that the FPT response is influenced by IR laser-induced excitation of intracellular molecules other than FPT, thermophoresis, or factors other than temperature (e.g., viscosity). The temperature selectivity of the FPT response demonstrated in these control experiments is supported by the results of Supplementary Figure 4. When we examined in detail the fluorescence lifetime of FPT and its component changes³¹ reflecting the chemical environment (e.g., hydration, hydrogen bonding, and hydrophobicity), these properties during IR laser irradiation were identical to those during heating of the medium, where the energy input can be regarded as solely thermal.

These multifaceted control experiments indicate that changes in factors other than intracellular temperature, caused by heating within the physiological range, do not affect the FPT response.” (Supplementary Note 4 in the Supplementary Information of the revised manuscript)

Figure 1d. Relationship between temperature and the fast fluorescence lifetime (τ_{FAST}) of CP.

Figure 2c. Fluorescence lifetime images of CP without (Off) and with (On) IR laser irradiation (0.55 mW). The right graph indicates the profile of $\Delta\tau_{\text{FAST}}$ of the cell.

Figure 3b. Tracking of fast fluorescence lifetime changes of FPT (red) and CP (black) in cells when heated by continuous (0.55 mW for 5 s) irradiation with an IR laser.

Supplementary Figure S5 | Fluorescence decay curve and τ_{FAST} distribution of CP in the IR laser irradiation region. **a–d** Analysis of fluorescence lifetimes (τ_1 and τ_2) and their compositions of fluorescence decay curves of CP without (**a,c**) and with IR laser irradiation (0.55 mW; **b,d**) in the cytoplasm (**a,b**) and in the nucleus (**c,d**). **e** The τ_{FAST} distribution of CP during IR laser irradiation (0.55 mW) in the cytoplasm and in the nucleus. The arrowheads indicate the point of IR laser irradiation. Scale bar represents 10 μm .

Response (continued): The degree of burial of probes serves as an indicator of molecules that interact with specific structures. Therefore, it does not apply to FPT, which is in a state of free diffusion within cells. To clarify this, we cited a paper (Ref. 2 in the revised manuscript) that investigated the diffusion state of intracellular FPT, and added an explanation to the revised manuscript:

Texts added to the revised manuscript (Results section):

“Intracellularly introduced FPT exists in a dispersed state within cells and can capture the temperature distribution in cells². This ability is advantageous over other intracellular temperature measurement methods, such as fixed-point temperature measurement using fluorescent nanoparticles or temperature tracking within specific organelles using fluorescent small molecules and proteins.” (Line 90-93 in the revised manuscript)

Comment: In Figure 1c, the authors compared the relationships between the fluorescence lifetimes of FPT in cells calculated by different methods (conventional method and fast method) and the temperature of the cellular medium. This is confusing because the authors aim to detect intracellular temperature through the probe's lifetime—that is, intracellular temperature is supposed to be calculated from the probe's lifetime. What is the purpose of analyzing their correlation?

Response: We deeply appreciate your insightful comment. As shown in Supplementary Note 1 of the previous manuscript, we presented two types of calibration curves for FPT to estimate the intracellular temperature. The calibration curve based on the medium temperature, as shown in Figure 1, was used solely to determine the intracellular temperature change (ΔT) during heating by an IR laser. This was necessary in order to ignore the temperature increase due to the steady-state spontaneous heat generation of cells. To avoid confusion, we replaced it with the calibration curve obtained from a solution (cell extracts; previously included in the Supplementary Information of the previous manuscript), a method already established for temperature mapping using FPT.

Texts added to the revised manuscript (Results section):

“Figure 1c shows the relationship between the temperature and the fast lifetime of FPT in the solution of COS7 cell extracts.” (Line 110-111 in the revised manuscript)

Comment: The authors' method for verifying the fast lifetime calculation (Figure 1d) is also highly confusing. Since the traditional method requires exponential function fitting, it needs more photons to obtain a convergent result, so the standard deviation (SD) must be large when the number of photons is small. In contrast, the authors calculate errors using an average value in a linear manner, resulting in a smaller SD. Therefore, the conclusion "As shown in Figure 1d, the determination of the fluorescence lifetime of FPT by fast lifetime achieved similar accuracy with only

4% of photons compared to the conventional method" is too arbitrary.

Response: We deeply appreciate your insightful comments. As Reviewer 1 pointed out, changing the function does not constitute the development of a new method. Therefore, we removed the following sentence from the revised manuscript: “As shown in Figure 1d, the determination of the fluorescence lifetime of FPT by fast lifetime achieved similar accuracy with only 4% of photons compared to the conventional method.” To comprehensively demonstrate the effect of the modified function on the performance of the temperature measurement, we included only the data for the analytical parameters (trueness and precision) that validate both the conventional and new methods.

Texts and figures added to the revised manuscript (Results section):

“Because we expected linear approximation to reduce variability in fluorescence lifetime determination, we adopted the fast fluorescence lifetime²⁸ defined by the average photon arrival time (“fast lifetime”), which requires fewer photons than the conventional method (Figure 1b, Supplementary Note 1).” (Line 106-108 in the revised manuscript)

“We investigated whether reliable fast lifetime measurements could be achieved using fewer photons than with conventional methods. We experimentally evaluated the impact of different fluorescence lifetime estimation algorithms on the trueness and precision of the resulting values. (Figures 1e, 1f, and Supplementary Figure 2). The results showed that fast lifetime-based fluorescence lifetime determination outperformed conventional methods in terms of trueness (Figure 1e) and precision (Figure 1f and Supplementary Figure 2). Furthermore, it was confirmed that the errors and variability associated with FPT fast lifetime determination using a limited number of photons did not significantly affect the measurement values in this study (Supplementary Note 1).” (Line 113-119 in the revised manuscript)

Figure 1e. Relationship between the accumulated photon count and fluorescence lifetime at 24, 27, and 30 °C (τ_f : left and τ_{FAST} : right).

Figure 1f. The relationship between photon counts in TCSPC-FLIM and the standard deviation of fluorescence lifetime (τ_f : black and τ_{FAST} : red).

Supplementary Figure S2 | Comparison of photon count-dependent fluorescence lifetime (τ_f vs. τ_{FAST}) variation. The fluorescence lifetime images of FPT were repeatedly obtained ten times and the SD of fluorescence lifetime (τ_f and τ_{FAST}) of a single living COS7 cell was calculated with various accumulating durations corresponding to acquired photon counts. Two kinds of fluorescence lifetime (τ_f and τ_{FAST}) at 24 °C and 30 °C are shown.

Comment: In this work, the authors irradiated living COS7 cells with an IR laser and studied intracellular temperature. How can they ensure that the laser affects only temperature rather than the structural characteristics or physiological functions of living cells, such as the enrichment of substances in intracellular regions? Which might also contribute to the fluorescent probes.

Response: We deeply appreciate your insightful comments. We verified that our conclusions were affected by the possibility that (a) FPT responds to factors other than temperature during IR laser irradiation and/or (b) IR laser irradiation induces changes in molecular structure, distribution, or reactions (including physiological functions) without involving heat. The results and discussions of the additional experiments were summarized in Supplementary Note 4, and an explanation was added to the revised manuscript.

Texts and figures added to the revised manuscript (Results, Methods sections, and Supplementary Note 4):

“The near-infrared spectrum of water showed absorption of light at the wavelength of the IR laser used in this study (1,480 nm), while no absorption of this light by FPT itself was observed (Supplementary Figure 3).” (Line

142-144 in the revised manuscript)

“Absorption spectra of FPT

Absorption spectra of FPT (0.1 w/v% in KCl aqueous solution [150 mM]) were obtained using an IRT-5200 Infrared Microscope (JASCO, Tokyo, Japan). A sample fixture was installed in a handmade cryostage for the microscope, and the sample temperature was stably controlled at 25 °C using liquid nitrogen and an electric heater integrated into the cryostage⁴⁶.” (Line 485-489 in the revised manuscript)

“Supplementary Note 4. Confirmation that the temperature relaxation after stopping IR laser irradiation is due to heat dissipation

In this work, we assumed that IR laser irradiation creates a heat source within cells. We confirmed no changes that differed from this assumption when examining relaxation using this approach were detected. We here verified that our conclusions were affected by the possibility that **(a)** FPT responds to factors other than temperature during IR laser irradiation and/or **(b)** IR laser irradiation induces changes in molecular structure, distribution, or reactions (including physiological functions) without involving heat.

(a) Temperature selectivity of the FPT response during IR laser irradiation

During heating, factors other than temperature, such as viscosity, also change. Previously, we confirmed that the fluorescence response of FPT is temperature-selective and remains unaffected by factors such as pH, ionic strength, biomolecule concentration, and viscosity². In this study, we investigated the temperature selectivity of the FPT response during IR laser irradiation

First, because FPT does not directly absorb IR laser light (Supplementary Figure 3), its response is independent of changes due to electronic excitation. Furthermore, CP having the same chemical composition as FPT but lacking temperature responsiveness did not respond to IR laser irradiation (Figures 2c, 3b, and Supplementary Figure 5). These results rule out the possibility that the FPT response is influenced by IR laser-induced excitation of intracellular molecules other than FPT, thermophoresis, or factors other than temperature (e.g., viscosity). The temperature selectivity of the FPT response demonstrated in these control experiments is supported by the results of Supplementary Figure 4. When we examined in detail the fluorescence lifetime of FPT and its component changes³¹ reflecting the chemical environment (e.g., hydration, hydrogen bonding, and hydrophobicity), these properties during IR laser irradiation were identical to those during heating of the medium, where the energy input can be regarded as solely thermal.

These multifaceted control experiments indicate that changes in factors other than intracellular temperature, caused by heating within the physiological range, do not affect the FPT response.

(b) Confirmation that the source of energy creating slow temperature relaxation is heat from IR laser

First, we recorded the intracellular temperature relaxation when only heat was transferred to the cells by irradiating the extracellular medium with an IR laser. The results showed that intracellular temperature relaxation was significantly slower than that within liposomes (Supplementary Figure 17a), similar to the findings for intracellular heating (Figure 3f). This demonstrated that the intracellular temperature changes that we observed were due to heat delivered by IR laser.

Furthermore, the emergence of other exothermic reactions (e.g., metabolic heat production) induced by IR laser irradiation might influence the interpretation of temperature relaxation. Therefore, we verified that intracellular temperature relaxation after stopping IR laser irradiation is affected by intrinsic cellular heat production, including physiological functions. The results showed that slow temperature relaxation occurred even in cells with inhibited metabolism, thereby depleting energy and halting intrinsic cellular heat production (Supplementary Figure 17b). The results indicate that the intracellular temperature relaxation observed is due to heat dissipation rather than a time-dependent decrease in intrinsic heat production.

These experimental verifications from various perspectives confirm that temperature relaxation after stopping IR laser irradiation is due to heat dissipation.” (Supplementary Note 4 in the Supplementary Information of the revised manuscript)

Supplementary Figure S3 | Near-infrared absorption spectra of FPT solution. The near-infrared absorption spectra of a KCl solution alone (150 mM) and that containing FPT (0.1 w/v%). The absorbance at 1480 nm (A_{1480}) is shown in the figures.

Figure 2c. Fluorescence lifetime images of CP without (Off) and with (On) IR laser irradiation (0.55 mW). The right graph indicates the profile of $\Delta\tau_{FAST}$ of the cell.

Figure 3b. Tracking of fast fluorescence lifetime changes of FPT (red) and CP (black) in cells when heated by continuous (0.55 mW for 5 s) irradiation with an IR laser.

Supplementary Figure S5 | Fluorescence decay curve and τ_{FAST} distribution of CP in the IR laser irradiation region. a–d Analysis of fluorescence lifetimes (τ_1 and τ_2) and their compositions of fluorescence decay curves of CP without (**a,c**) and with IR laser irradiation (0.55 mW; **b,d**) in the cytoplasm (**a,b**) and in the nucleus (**c,d**). **e** The τ_{FAST} distribution of CP during IR laser irradiation (0.55 mW) in the cytoplasm and in the nucleus. The arrowheads indicate the point of IR laser irradiation. Scale bar represents 10 μm .

Supplementary Figure S17 | The temperature relaxation rate in living cells when heating the

extracellular space and when metabolic heat production is inhibited. a,b Comparison of the normalized temperature relaxation in cells after stopping heating of the extracellular medium (IR laser: 0.83 mW for 5 s, red; **a**) and of metabolic heat-inhibited cells (IR laser: 0.55 mW for 5 s, red; **b**), and the normalized temperature relaxation rate within liposomes (black, Figure 3f).

Comment: In addition to the doubtful conclusions, the paper's innovativeness is also limited. Merely changing the calculation method for fluorescence lifetime lacks theoretical basis or innovation and fails to provide new mechanistic solutions.

Response: We greatly appreciate your feedback. We firmly believe that the innovativeness of this paper lies not in the modification of the method for calculating fluorescence lifetime, but rather in the elucidated mechanism of intracellular heat dissipation. In the revised manuscript, we have therefore modified the title, abstract, introduction, and discussion sections to clearly state that the innovation of this paper is the scientific discovery made by examining the mechanism of intracellular heat dissipation, not the methodological modification.

Texts added to the revised manuscript (Title, Abstract, Introduction, and Discussion sections):

Title

“Non-Diffusive Slow Heat Dissipation Induces High Local Temperature in Living Cells”

Abstract

“Here, we investigate intracellular heat transfer through intracellular temperature mapping using a fluorescent polymeric thermometer and high-speed fluorescence lifetime imaging microscopy. Through infrared laser irradiation-assisted heating, we track changes in temperature distribution to examine the mechanism of intracellular heat dissipation in comparison with heat conduction. Continuous heating induces the significantly slower relaxation of the average temperature of single cells compared with that of liposomes containing homogeneous aqueous solutions of comparable size; to the scale of seconds. We additionally elucidate that these phenomena are impacted by intracellular structures and molecules. Lastly, we discover a fundamentally new form of intracellular temperature relaxation that is non-diffusive and independent of heat conduction.” (Line 21-

Introduction

“It is well established that physiological functions include adapting to and utilizing changes in ambient temperature. The study of the control of biological functions through heating has a long history, under topics such as hyperthermia therapy in medicine and the heat shock response in biology. In recent years, intracellular temperature has attracted increasing attention. Fluorescent molecular thermometers (fluorescent polymeric thermometers [FPT]^{1,2}, fluorescent proteins³⁻⁵, fluorescent small molecules⁶, and nanoparticles such as upconversion nanoparticles [UCNP]^{7,8} and fluorescent nanodiamonds [FND]⁹⁻¹¹) and non-optical methods (thermocouples^{12,13} and electrochemical methods¹⁴) have revealed temporal and spatial variations of temperature within cells. These intracellular temperature changes are characterized by spontaneous heat generation within cells and are associated with cellular functions¹⁵⁻¹⁸.” (Line 34-42 in the revised manuscript)

“This novel phenomenon produced by intracellular heat propagation (termed thermal signaling) is considered to serve as a universal intracellular signaling mechanism as all molecular dynamics and chemical reactions are governed by heat. Thus, intrinsic temperature changes in cells are a promising novel property in biology and medicine that drives cellular activity and determines the cellular state¹⁵⁻¹⁸.”

Although the biological significance of intracellular temperature variations is being revealed, the mechanism by which heat generated within the cell produces this temperature change of approximately 1–2 °C remains unknown. It has been pointed out that, according to the heat conduction equation, temperature changes due to spontaneous heat generation within cells can only reach approximately 10⁻⁵ °C (the 10⁵ gap issue)²¹. While the values calculated from the heat conduction equation and experimental observations show a significant discrepancy, the observed result of intracellular temperature change by about 1 °C due to spontaneous heat generation has been verified through robust reproducibility, across different methodological principles¹⁵. Via attempts to explain the physical mechanism behind this discrepancy, it has been proposed that heat generated within intracellular regions where heat conduction is restricted could lead to elevated temperatures^{5,22,23}. Recently, various studies have estimated cellular thermal conductivity (0.1–0.6 W m⁻¹ K⁻¹)²³⁻²⁶ and thermal diffusivity (0.27–1.34×10⁻⁷ m² s⁻¹)^{5,27} to be between one-sixth and one-fold of those in water. It should be noted that the actual thermal conductivity and diffusivity reported in these studies showed significant variation. Even assuming conditions in which heat conduction is severely restricted (thermal conductivity: 0.1 W m⁻¹ K⁻¹, thermal diffusivity: 0.27×10⁻⁷ m² s⁻¹), the discrepancy between the experimental results and the calculated values remains substantial (over 10⁴-fold). This gap cannot be fully explained by considering additional parameters, such as the intensity of and the distance from the heat source. Thus, this issue concerning intracellular temperature variation is not a matter of the cellular thermal properties, but rather a fundamental mystery of the heat dissipation process itself¹⁷.” (Line 46-67 in the revised manuscript)

Discussion

“Additionally, by exploiting the high temporal and spatial resolution of temperature mapping, we observed transient changes in intracellular temperature distribution after stopping heating and discovered that slow and non-diffusive heat dissipation within cells does not depend on heat conduction.” (Line 303-305 in the revised manuscript)

Comment: The language in the paper is not well organized and optimized. For example: "FLIM using the most accurate and pervasive TCSPC was used for detecting the fluorescence lifetime of intracellular FPT."

Response: We appreciate your feedback. We reviewed our paper and revised the suboptimal language. The sentences you pointed out were deleted during the revision process.

Comment: The formatting of formulas in the methods section needs revision, and the formulas and the parameters could be separated.

Response: We appreciate your feedback. The formulas and parameters have been separated, with the parameters moved to the Supplementary Information.

Texts and table added to the revised manuscript (Methods section):

$$\tau_{\text{FAST}}(T) = A_1T^6 + A_2T^5 + A_3T^4 + A_4T^3 + A_5T^2 + A_6T + A_7 \quad (1)$$

where T and $\tau_{\text{FAST}}(T)$ represent the temperature ($^{\circ}\text{C}$) and the fast fluorescence lifetime (ns) at $T^{\circ}\text{C}$, respectively. The coefficients (A_1 – A_7) in Equation (1) are listed in Supplementary Table.” (Line 455-458 in the revised manuscript)

Table. Coefficients in the polynomial of Equation (1)

	Living COS7 cell	COS7 cell extracts	Living HeLa cell
A_1	0	−0.0000046669	0.05049671042
A_2	0.0603644272864967	0.0008623426	−2.26638446940322
A_3	−2.27554925190688	−0.0654657569	42.0594063916615
A_4	34.1786499772982	2.6100824804	412.714691229774
A_5	−255.3666687	57.5778063053	2256.40968069216
A_6	949.582685485572	666.2704802512	−6509.58277053094
A_7	−1381.94789212487	−3157.2782188756	7755.40769741854

Comment: The Introduction section is overly simplistic, failing to fully elaborate on the background and current research progress. Some key literature is also not cited.

Response: We appreciate your feedback. In response to this and other comments, we have added a detailed description of the background and the current state of research, along with key references, to the introduction section. We revised the introduction in conjunction with your comments regarding the novelty of the research. Please refer to the aforementioned response. We have added the following new references:

References added to the revised manuscript (Reference section):

10. Petrini, G. *et al.* Nanodiamond–quantum sensors reveal temperature variation associated to hippocampal neurons firing. *Adv. Sci.* **9**(28), 2202014. (2022).
11. Lee, Y., Kim, K., Kim, D., & Lee, J. S. Organelle-Specific Quantum Thermometry Using Fluorescent Nanodiamonds: Insights into Cellular Metabolic Thermodynamics. *J. Am. Chem. Soc.* **147**(16), 13180-13189 (2025).
13. Wu, L. *et al.* Combined cellular thermometry reveals that salmonella typhimurium warms macrophages by inducing a pyroptosis-like phenotype. *J. Am. Chem. Soc.* **144**(42), 19396-19409 (2022).
14. Ding, H., Liu, K., Zhao, X., Su, B., & Jiang, D. Thermoelectric nanofluidics probing thermal heterogeneity inside single cells. *J. Am. Chem. Soc.* **145**(41), 22433-22441 (2023).
22. Suzuki, M. & Plakhotnik, T. The challenge of intracellular temperature. *Biophys. Rev.* **12**(2), 593-600 (2020).
24. Song, P. *et al.* Heat transfer and thermoregulation within single cells revealed by transient plasmonic imaging. *Chem* **7**(6), 1569-1587 (2021).
25. Park, B. K. *et al.* Thermal conductivity of biological cells at cellular level and correlation with disease state. *J. Appl. Phys.* **119**(22) (2016).
26. Shrestha, R., Atluri, R., Simmons, D. P., Kim, D. S., & Choi, T. Y. Thermal conductivity of a Jurkat cell measured by a transient laser point heating method. *Int. J. Heat Mass Transf.* **160**, 120161 (2020).

Reviewer #2 (Remarks to the Author)

Comment: Takarada et al used fast FLIM imaging method to measure the dynamics heat dissipation inside cells. It was found that the heat dissipation inside cells is significantly longer than the time scale predicted by heat conduction theory. In terms of methodology, the fast FLIM method they used significantly improved the temporal resolution of fluorescent temperature measurement, enabling the tracking of the heat dissipation process inside cells on ms time scale, which is very impressive. From a scientific discovery perspective, the observation of such a long time-scale heat decay inside cells is surprising and exciting. This may imply that the thermodynamics inside cells do not conform to traditional theoretical models, if the experimental measurements are credible. This will undoubtedly update our understanding of the thermodynamics of living systems. Overall, I strongly recommend the publication of this work.

To enhance the strength of this work, I have the following suggestions:

Response: We deeply appreciate Reviewer #2's positive comments on our manuscript and constructive ideas for the revision. We were very impressed by the words of encouragement.

Comment: The author's team is one of the pioneers in the use of fluorescence thermometry to measure cell temperature and has carried out a series of exploratory work on cell thermodynamics. The FPT they developed is also a very typical fluorescence thermometer. I appreciate these contributions. Recently, the measurement uncertainty of fluorescence thermometers has been noticed. Especially for small-molecule fluorescence thermometers, how to ensure that the measured spectral changes (such as lifetime) depend solely on temperature but not other factors? This seems difficult because, from the principles of photo-physical chemistry, in addition to temperature, the viscosity of the medium is also an important factor in fluorescence lifetime. Can the authors comment on this issue? For example, is the calibration curve shown in Figure 1 carried out in solution or in cells? How they fix this issue on their FPT?

Response: We deeply appreciate your kind words of encouragement and insightful comments on our previous research. When developing FPT, we prioritized ensuring that its fluorescence lifetime response was derived solely from temperature and unaffected by chemical, physical, or biological factors. In our previous work, we demonstrated that FPT selectively detects intracellular temperature changes due to its functional independence; FPT did not respond to physicochemical factors other than temperature, such as pH, ionic strength, biomolecular concentration, and viscosity, when they were quantitatively altered in solution (Ref. 2 in the revised manuscript). Furthermore, we used a control copolymer (CP) with a chemical composition almost identical to that of FPT, but lacking temperature sensitivity, to confirm the temperature selectivity of FPT. In complex environments like the intracellular space, various physicochemical changes can occur due to heating. Therefore, rigorous verification using this CP is essential to achieving our objectives. The inclusion of a control probe that enables the discussion of temperature selectivity is a unique feature of FPT that is not found in other nanothermometers. Thus, we added an explanation of this functional independence of FPT to the manuscript and investigated the response of CP during intracellular heating. Through various investigations using CP, we demonstrated that FPT is unaffected by changes in medium viscosity, which may vary during heating. This strong evidence that the intracellular FPT response is due solely to temperature changes strengthens our claim in this paper; thus, we added this result to the revised manuscript.

The calibration curve shown in Figure 1 was obtained from living cells. We originally used this temperature response curve of τ_{FAST} of FPT obtained in living cells to determine the intracellular temperature change (ΔT) during heating by an IR laser. This was necessary in order to ignore the temperature increase due to the steady-state spontaneous heat generation of cells. For intracellular temperature mapping, we used a previously established calibration method using cell extracts (Figure 1c). To avoid confusion, we replaced this with the calibration curve obtained from a solution (cell extracts; previously included in the Supplementary Information of the previous manuscript), a method already established for temperature mapping using FPT.

Texts and figures added to the revised manuscript (Results section):

“Intracellularly introduced FPT exists in a dispersed state within cells and can capture the temperature distribution in cells². This ability is advantageous over other intracellular temperature measurement methods, such as fixed-point temperature measurement using fluorescent nanoparticles or temperature tracking within specific organelles using fluorescent small molecules and proteins. We previously demonstrated that FPT selectively detects intracellular temperature changes due to its functional independence; FPT did not respond to physicochemical factors other than temperature, such as pH, ionic strength, biomolecular concentration, and viscosity, when they were quantitatively altered in solution². In this study, we used a control copolymer (CP)² with a chemical composition almost identical to that of FPT, but lacking temperature sensitivity, to confirm the temperature selectivity of FPT. In complex environments like the intracellular space, various physicochemical changes can occur due to heating. Therefore, rigorous verification using this CP is essential to achieving our objectives. The inclusion of a control probe that enables the investigation of temperature selectivity is a unique feature of FPT that is not found in other nanothermometers.” (Line 90-101 in the revised manuscript)

“We confirmed that CP does not exhibit temperature dependence even in fast lifetime measurements (Figure 1d).” (Line 113-114 in the revised manuscript)

“Furthermore, the analysis of the fluorescence decay curve of CP confirmed that factors other than temperature (e.g., viscosity) induced by IR laser irradiation do not directly affect the FPT response (Supplementary Figure 5).” (Line 149-151 in the revised manuscript)

“On the other hand, IR laser irradiation did not change the fluorescence lifetime distribution of CP (Figure 2c). These results clearly demonstrate that the spatial distribution of the fluorescence lifetime of FPT created by IR laser irradiation is solely due to heat, independent of the chemical environment.” (Line 153-156 in the revised manuscript)

“Using CP, we confirmed that FPT responds only to temperature changes induced by IR laser irradiation (Figure 3b).” (Line 183-184 in the revised manuscript)

“Based on the temperature selectivity and fast response time of FPT, the results of control experiments using CP, the independence of the measurement methodology, and the fast relaxation at specific heating durations (e.g., 100 ms), we concluded that this significantly slower temperature relaxation measured inside cells compared to a homogeneous aqueous solution when cells were continuously heated is an intrinsic property of cells, and is not due to undesirable side effects of FPT.” (Line 298-302 in the revised manuscript)

Figure 1d. Relationship between temperature and the fast fluorescence lifetime (τ_{FAST}) of the CP.

Figure 2c. Fluorescence lifetime images of CP without (Off) and with (On) IR laser irradiation (0.55 mW). The right graph indicates the profile of $\Delta\tau_{FAST}$ of the cell.

Figure 3b. Tracking of fast fluorescence lifetime changes of FPT (red) and CP (black) in cells when heated by continuous (0.55 mW for 5 s) irradiation with an IR laser.

Supplementary Figure S5 | Fluorescence decay curve and τ_{FAST} distribution of CP in the IR laser irradiation region. a–d Analysis of fluorescence lifetimes (τ_1 and τ_2) and their compositions of fluorescence decay curves of CP without (a,c) and with IR laser irradiation (0.55 mW; b,d) in the cytoplasm (a,b) and in the nucleus (c,d). **e** The τ_{FAST} distribution of CP during IR laser irradiation (0.55 mW) in the cytoplasm and in the nucleus. The arrowheads indicate the point of IR laser irradiation. Scale bar represents 10 μm .

Comment: Are there any control experiments in the medium to see how long the heat dissipation time scale is in known media and whether it is related to the thermal conductivity and thermal diffusion of the medium? These control experiments are very important. On the one hand, it can argue for the effectiveness of the method, and on the other hand, it can also compare with the situation inside cells. I strongly suggest the authors to try heat dissipation tracking experiments in water and glycerol at different proportions to see the time scale of heat dissipation, and how the timescale related to their heat conductivity? There will definitely be interesting results.

Response: We deeply appreciate your comments. First, the original manuscript included a description of a control experiment in a homogeneous medium against the observation of heat dissipation within cells. In this control experiment, the temperature relaxation time in water encapsulated in liposomes was significantly faster than that

within cells (Figure 3f in the revised manuscript), clearly demonstrating that the slow temperature relaxation within cells is not due to methodological inadequacy.

In the previous manuscript, we did not consider the influence of the thermal conductivity of cellular components, including glycerol, as was suggested. This is because the conclusions of this paper involve phenomena with distinct timescales from those of heat conduction-based temperature relaxation. Following this comment, we examined how differences in thermal conductivity influence the rate of thermal relaxation. By comparing numerical calculations of thermal relaxation due to heat conduction in proteins, which have low thermal conductivity among biomolecules, with calculations in water, we confirmed that the thermal relaxation time for proteins is longer than in water. However, the relaxation time was still on the order of milliseconds. It should be noted that, because this timescale (approximately 1 ms) is shorter than the time required for structural changes in FPT (13 ms; Supplementary Figure 8), we concluded that this temperature tracking method cannot detect slight differences in thermal relaxation due to heat conduction. Thus, the following changes have been made in the revised manuscript:

Texts and figure added to the revised manuscript (Supplementary Note 5):

“Using this equation (S7) and the values of thermal conductivity (λ : 0.618 W m⁻¹ K⁻¹), specific heat capacity (c : 4.178 J g⁻¹ K⁻¹) and density (ρ : 1 g mL⁻¹) of water, and the radius of heat source (a : 0.65 μ m), we numerically calculated the time course of temperature. As shown in Supplementary Figure 18, the relaxation time was on the order of μ s–ms. Meanwhile, because various biomolecules (e.g., proteins, lipids, and glycerol) are present in cells, we examined how their thermal properties, particularly their low thermal conductivity, influence the time required for temperature relaxation. We numerically calculated the time course of temperature by using the values of thermal conductivity (λ : 0.1 W m⁻¹ K⁻¹)^{49,50}, specific heat capacity (c : 3.9 J g⁻¹ K⁻¹)⁵¹ and density (ρ : 1.05 g mL⁻¹)⁵² of cellular components. As shown in Supplementary Figure 18, these comparisons revealed that, although the low thermal conductivity of cellular components slows temperature relaxation, the relaxation due to heat conduction still occurs on the order of ms. It should be noted that the temporal resolution of this temperature tracking method (9 ms) appears insufficient for detecting subtle differences in temperature relaxation caused by variations in the thermal properties of intracellular components.” (Supplementary Note 5 in the Supplementary Information of the revised manuscript)

Supplementary Figure S18 | Time-course of normalized temperature relaxation after stopping heating numerically simulated with the heat conduction equation. The relaxations of average temperature of an area of a 5 μm radius centered on the heating point. The numerical simulations were performed with the heat conduction equation (S3) and the thermal properties reported for water (black) and cell components (red) (Supplementary Note 5).

Comment: I have a technical concern about fast FLIM. Apparently, to increase the time resolution, the accumulation of photons is reduced, as shown in the methods section of the paper. With the decrease in photons, shot noise will increase significantly, which in turn will increase the error in lifetime fitting. Has the author quantitatively assessed the error range of fast FLIM, in principle and in theory, and is it within an acceptable level? Of course, the present data shows some comparisons of individual curves and they fit pretty wells, but that is not enough to convince me completely.

Response: We deeply appreciate your insightful comments. Originally, we performed high-speed tracking of temperature (distribution) using the fast lifetime of the FPT under measurement conditions (e.g., laser intensity, scan speed, and accumulation time) that maintain the accuracy of fast lifetime determination. Following your suggestion, we confirmed that potential errors arising from variability in high-speed lifetime determination under limited photon availability and dark noise are acceptable.

First, the photon count obtained in the most photon-limited high-resolution temperature measurement conducted in this study was $16,967 \pm 4,209$ ($n = 30$). In contrast, the temperature response curve for fast lifetime measurements (Figure 1c) indicates that approximately 10,000 photons are required to determine τ_{FAST} with variance below the temperature resolution of about 0.2 °C (corresponding to a change of about 0.1 ns in the S.D. value of the fast lifetime; Figure 1f). Thus, the variability in FPT fast lifetime determination does not affect the measurements in this study. Second, the number of photons originating from dark noise in the detector is 8.7 ± 1.7 ($n = 10$), which gives an error to τ_{FAST} of approximately 0.01 ns. This indicates that the error introduced by dark noise is negligible.

We have added this consideration to the main text and Supplementary Note 1, as shown below.

Texts and figures added to the revised manuscript (Results sections and Supplementary Note 1):

“We investigated whether reliable fast lifetime measurements could be achieved using fewer photons than with conventional methods. We experimentally evaluated the impact of different fluorescence lifetime estimation algorithms on the trueness and precision of the resulting values. (Figures 1e, 1f, and Supplementary Figure 2). The results showed that fast lifetime-based fluorescence lifetime determination outperformed conventional methods in terms of trueness (Figure 1e) and precision (Figure 1f and Supplementary Figure 2). Furthermore, it was confirmed that the errors and variability associated with FPT fast lifetime determination using a limited number of photons did not significantly affect the measurement values in this study (Supplementary Note 1).” (Line 115-121 in the revised manuscript)

“We investigated the acceptable variability in fast lifetime determination under limited photon availability when measuring small intracellular temperature changes (with a temperature resolution of approximately $0.2\text{ }^{\circ}\text{C}$)² using FPT. First, the photon count obtained in the most photon-limited high-resolution temperature measurement conducted in this study was $16,967 \pm 4,209$ ($n = 30$). In contrast, the temperature response curve for fast lifetime measurements (Figure 1c) indicates that approximately 10,000 photons are required to determine τ_{FAST} with variance below the temperature resolution of about $0.2\text{ }^{\circ}\text{C}$ (corresponding to a change of about 0.1 ns in the S.D. value of the fast lifetime; Figure 1f). Thus, the variability in FPT fast lifetime determination does not affect the measurements in this study. Second, the number of photons originating from dark noise in the detector is 8.7 ± 1.7 ($n = 10$), which gives an error to τ_{FAST} of approximately 0.01 ns. This indicates that the error introduced by dark noise is negligible.” (Supplementary Note 1 in the Supplementary Information of the revised manuscript)

Figure 1e. Relationship between the accumulated photon count and fluorescence lifetime at 24, 27, and 30 °C (τ_f : left and τ_{FAST} : right). **f.** The relationship between photon counts in TCSPC-FLIM and the standard deviation of fluorescence lifetime (τ_f : black and τ_{FAST} : red).

Figure 1f. The relationship between photon counts in TCSPC-FLIM and the standard deviation of fluorescence lifetime (τ_f : black and τ_{FAST} : red).

Supplementary Figure S2 | Comparison of photon count-dependent fluorescence lifetime (τ_f vs. τ_{FAST}) variation. The fluorescence lifetime images of FPT were repeatedly obtained ten times and the SD of fluorescence lifetime (τ_f and τ_{FAST}) of a single living COS7 cell was calculated with various accumulating durations corresponding to acquired photon counts. Two kinds of fluorescence lifetime (τ_f and τ_{FAST}) at 24 °C and 30 °C are shown.

Comment: In addition to fluorescence thermometers, there are also other methods to try to measure the heat dissipation and thermal relaxation inside cells. For example, in the (Song et al. Chem, 2021, 7-6, 1569-1587) literature, a heated nanoprobe was used to observe a heat relaxation process in cells, and they also observed significant slower thermal decay than the timescale predicted by the heat conduction model (>two orders of magnitude). Although the methods differ, the main results seem point in the same direction. I suggest that the authors cite this literature and briefly discuss this point.

Response: We appreciate your suggestion. We have carefully examined the temperature changes and relaxations reported in the literature on heat transfer and dissipation within cells. The study by Song P. *et al.* (Ref. 24 in the revised manuscript) reports slow intracellular temperature relaxation. However, this method does not significantly deviate (only a few-fold difference) from the relaxation observed in water, where temperature relaxes due to heat conduction. Therefore, the mechanism of heat conduction-dependent temperature changes described in this paper is fundamentally different from our findings. In contrast, the phenomenon that we discovered accounts for the low intracellular thermal conductivity and thermal diffusivity estimated by molecular thermometers, including those of

Song *et al.* We have added this point to the discussion section of the revised manuscript.

Texts added to the revised manuscript (Discussion section):

“The cellular thermal conductivity recently estimated from the local temperature change of molecular thermometers^{5,23,24} differed significantly from the cellular thermal conductivity determined from a measurement in bulk (cell suspension) at a relatively large spatial scale²⁵ and equilibrium thermal measurements^{26,27}. The low thermal conductivities and the thermal diffusivity estimated in cells using the former methods should reflect the significant influence of molecular relaxation on temperature changes at the single-cell scale, as discovered in this study.” (Line 365-370 in the revised manuscript)

Comment: Regarding why there is such slow heat relaxation inside cells, in addition to the energy conversion possibilities discussed by the authors, I have some different views. In fact, the classical laws of heat transfer, including Fourier's law of heat conduction and Newton's law of cooling, were derived and verified in macroscopic, homogeneous media. For micro-nano scales with complex multiphase compositions, such as cells, the above theoretical models may show significant deviations, or even be completely inapplicable. In the case of cells, the complex organelles and locally concentrated biomacromolecules (phase separation) inside cells create complex multiphase interfaces. These interfaces create interfacial thermal resistance (Kapitza thermal resistance) for heat transfer. Perhaps the Kapitza thermal resistance on one interface is negligible, but there are thousands of such interfacial thermal resistances inside cells, which will significantly slow down the dissipation of heat. I suggest that the authors also consider this possibility, if the length of the discussion allows.

Response: We greatly appreciate your comments and suggestions. Following your suggestion, we examined the influence of these interfaces on intracellular heat dissipation rates, using interfacial thermal resistance values for lipid/protein–water interfaces obtained from experimental measurements or numerical analysis as a reference; we found that the presence of thermal resistance extends thermal relaxation times. However, we also confirmed that the slow thermal relaxation occurring on the order of seconds within the intracellular space cannot be explained by heat conduction alone, even when considering thermal resistance. This examination has been added to the revised manuscript as Supplementary Note 7.

Texts added to the revised manuscript (Discussion section and Supplementary Note 7):

“Furthermore, based on the discussion in Supplementary Note 7 regarding the possibility that thermal resistance at multiphase interfaces within the cell slows heat dissipation, the slow thermal relaxation occurring on the order of seconds within the intracellular space cannot be explained by heat conduction alone, even when considering thermal resistance.” (Line 310-313 in the revised manuscript)

“Therefore, the time required for heat dissipation via heat conduction including the effects of interfacial thermal resistance in spaces of certain sizes (i.e., in large spaces) is non-negligible compared with the relaxation time of the molecules.” (Line 357-359 in the revised manuscript)

“Supplementary Note 7. Effect of thermal resistance at intracellular multiphase interfaces on heat dissipation rate

Within cells, organelles, phase-separated condensates, and locally concentrated biomolecules form complex multiphase interfaces. The interfacial thermal resistance created by numerous interfaces of this kind may slow heat dissipation. We examined the influence of these interfaces on intracellular heat dissipation rates, using interfacial thermal resistance values for lipid/protein–water interfaces obtained from experimental measurements or numerical analysis as a reference.

We used thermal resistance values for interfaces created by biomacromolecules estimated from theoretical calculations and experiments in previous studies. The thermal resistance (R) of a single interface is approximately $1\text{--}5 \times 10^{-9} \text{ m}^2 \text{ K W}^{-1}$ (proteins: $3 \times 10^{-9} \text{ m}^2 \text{ K W}^{-1}$; lipids: $1\text{--}5 \times 10^{-9} \text{ m}^2 \text{ K W}^{-1}$)^{49,53,54}.

We assumed a situation in which the highest thermal resistance ($R = 5 \times 10^{-9} \text{ m}^2 \text{ K W}^{-1}$) and total thermal resistance (R_{total}) of 1,000 interface layers within a representative length of $L = 10 \text{ }\mu\text{m}$ hindered heat conduction within the cell. Considering the additional effect of interfacial thermal resistance on the intrinsic thermal conductivity ($\lambda = 0.1 \text{ W m}^{-1} \text{ K}^{-1}$) within the cell, the effective thermal conductivity (λ_{eff}) is:

$$R_{total} = \frac{L}{\lambda} + R \times 1000 = \frac{10 \times 10^{-6}}{0.1} + 5 \times 10^{-9} \times 1000 = 1.05 \times 10^{-4} \text{ m}^2 \text{ K W}^{-1}$$
$$\lambda_{eff} = \frac{L}{R_{total}} = \frac{10 \times 10^{-6}}{1.05 \times 10^{-4}} \approx 0.095 \text{ W m}^{-1} \text{ K}^{-1} \quad (\text{S8})$$

The presence of thermal resistance reduces the apparent thermal conductivity to 95% and extends the thermal relaxation time. However, the effect on thermal relaxation time is an increase of at most 5%, indicating that the order of magnitude remains unchanged (i.e., it significantly deviates from the measured temperature relaxation timescale in this experiment).” (Supplementary Note 7 in the Supplementary Information of the revised manuscript)

Comment: Some detailed errors, the content of Figure 4b seems inconsistent with the corresponding figure legend description (top, middle, bottom); There is an absence of a space between numerical values and the degree Celsius symbol; in certain expressions, such as “2.9 mgmL⁻¹”, which clearly requires standardization.

Response: We greatly appreciate your comments. The legend in Figure 4b and the unit expressions have been corrected.